# Structure of the *Methanosarcina mazei* Mtr complex bound to the oxygen-stress responsive small protein MtrI

Tristan Reif-Trauttmansdorff[1,5], Eva Herdering[2,5], Stefan Bohn [3], Tomas Pascoa[1], Jörg Kahnt[4], Erik Zimmer [1], Anuj Kumar[1], Ruth A. Schmitz [2] ✉ & Jan M. Schuller [1] ✉

Methanogenic archaea emit ~1 Gt of methane annually, impacting global carbon cycling and climate. Central to their energy metabolism is a membrane-bound, sodium-translocating methyltransferase complex: the $N^5$-tetrahydromethanopterin:CoM-S-methyltransferase (Mtr). It couples methyl transfer between two methanogen-specific cofactors with sodium ion transport across the membrane, forming the only energy-conserving step in hydrogenotrophic methanogenesis. Here, we present a 2.1 Å single-particle cryo-EM structure of the Mtr complex from *Methanosarcina mazei*. The structure reveals the organization of all catalytic subunits, embedded archaeal lipids and the sodium-binding site. Most strikingly, we discover MtrI, a previously unannotated small open-reading frame encoded protein (<100 aa) found within the order of Methanosarcinales that binds both the top of the sodium-channel and cytosolic domain of MtrA via its cobamide cofactor in response to oxygen exposure. This interaction likely prevents sodium leakage and stabilizes the complex under oxidative conditions, revealing an unexpected regulatory mechanism in methanogen energy conservation.

Methanogenic archaea are widespread in different environments with high abundancies growing under strictly anoxic conditions and are crucial for the last step in anaerobic degradation of organic matter. They produce methane as a primary catabolic end product of their energy metabolism, although methane can also be formed in trace amounts by a variety of other organisms via alternative biochemical pathways[1–5]. Consequently, they produce approximately one gigaton of this greenhouse gas per year[6]. Methanogenesis occurs via several pathways using different substrates: e.g., hydrogenotrophic methanogenesis ($H_2$/$CO_2$), acetoclastic methanogenesis (acetate) and methylotrophic methanogenesis (e.g., methylamines and methanol)[7,8]. A pivotal step in all methanogenic routes is catalysed by the membrane-bound $N^5$-methyltetrahydromethanopterin:coenzyme M methyltransferase complex (Mtr), which mediates the exergonic transfer of a methyl group from methyl-tetrahydromethanopterin (methyl-$H_4$MPT) or methyl-tetrahydrosarcinapterin ($H_4$SPT in some species) to HS-CoM ($\Delta G^{o\prime} = -30$ kJ/mol), coupled to sodium ion translocation across the membrane[9–12]. This sodium-motive force drives chemiosmotic energy conservation and ATP synthesis via a Na$^+$-dependent ATP synthase, forming the basis of what is often referred to as sodium-based bioenergetics in methanogenic archaea[13–15]. Methylotrophic methanogens convert methylated compounds like methanol or methylamines to methane and $CO_2$ through a disproportionation process. For example, one molecule of methanol is oxidised to $CO_2$,

[1]Center for Synthetic Microbiology (SYNMIKRO) Research Center and Department of Chemistry, Philipps-Universität Marburg, Karl-von-Frisch Straße 14, Marburg, Germany. [2]Institute for General Microbiology, Christian Albrechts University, Am Botanischen Garten 1-9, Kiel, Germany. [3]Helmholtz Munich Cryo-Electron Microscopy Platform, Institute of Structural Biology, Helmholtz Munich, Ingolstädter Landstrasse 1, Neuherberg, Germany. [4]Max Planck Institute for Terrestrial Microbiology and Department of Biology, Philipps-University Marburg, Marburg, Germany. [5]These authors contributed equally: Tristan Reif-Trauttmansdorff, Eva Herdering. ✉e-mail: rschmitz@ifam.uni-kiel.de; jan.schuller@synmikro.uni-marburg.de

and the released electrons are used to reduce three additional methanol molecules to methane. Under these conditions, the Mtr reaction is reversed, and the sodium ion gradient is used to drive the endergonic methyl-transfer from methyl-CoM to H$_4$MPT. Similarly, anaerobic methanotrophic archaea (ANME) oxidise methane to CO$_2$ by operating the methanogenesis pathway, and thus the Mtr enzyme, entirely in reverse[16].

Mtr complexes in methanoarchaea have been studied by biochemical and genetic approaches for more than thirty years[17]. It has been known for a long time that Mtr complexes consist of eight subunits (MtrABCDEFGH), of which the corresponding genes are organised in an operon, which is constitutively expressed[18]. The full Mtr complex was found to have an apparent molecular mass of about 650 kDa, concluding a heterotrimeric complex (MtrABCDEFGH)$_3$[19]. This heterotrimeric complex consists of the cytosolic catalytic domains and a large membrane portion including the sodium channel. Thereby, MtrC, D and E are the largest integral membrane subunits, MtrA, B, F and G contain membrane anchors, and MtrH is the soluble component in the cytosol[17].

Structural data on Mtr is currently limited to a partial cryo-EM reconstruction of the membrane subunits and an X-ray structure of the MtrA shuttle domain[20,21], excluding the cytosolic MtrH and full-length MtrA components. While incomplete, these structures provide key insights– most notably the identification of a sodium-binding site in the MtrE subunit involving the conserved Asp187, previously implicated in sodium ion transport[17]. A second putative Na$^+$ binding site and a coenzyme M (CoM) molecule were tentatively assigned within the central MtrC cavity, though these remain to be functionally validated[20]. In addition, the crystal structure of MtrA revealed a unique corrinoid-binding mode, distinct from known B12-binding proteins, utilising a Rossmann-fold domain[21]. However, the absence of a full complex structure in a catalytically competent state continues to limit mechanistic understanding of sodium-coupled methyl transfer.

While much focus has been on mechanistic studies to elucidate the mechanism of sodium translocation, relatively little attention has been paid to the regulation and integration of Mtr within the broader cellular context. In particular, the mechanisms governing Mtr regulation, as well as its potential modulation in response to environmental cues, are poorly understood.

Recent advances in ribosome profiling and quantitative proteomics have revealed an additional regulatory layer in prokaryotes, mediated by small proteins encoded by small open reading frames (sORFs)[22,23]. Defined as polypeptides typically shorter than 100 amino acids, these proteins are frequently omitted from genome annotations due to size thresholds, yet have emerged as widespread and functionally diverse regulators across bacteria and archaea. Expression profiling showed that many small proteins accumulate under specific growth conditions or are induced by stress arguing for a role in regulation[24]. However, information regarding a specific physiological role of verified small proteins is frequently lacking for the majority of confirmed small proteins. For *Methanosarcina mazei*, we recently reported on 314 previously non-annotated small ORF-encoded small proteins, several of them regulated in response to nitrogen stress[23]. Although their precise roles remain to be established, the observed condition-dependent expression patterns raise the possibility that some may modulate key metabolic processes, including Mtr complex function or responses to environmental stressors.

In this study, we present the single-particle cryo-electron microscopy (cryo-EM) structure of the complete Mtr complex from *M. mazei*, obtained at a resolution of 2.1 Å without applying any symmetry (C1). Our results detail the overall architecture of the complex, including tightly bound archaeal phospholipids and the interaction of the MtrH methyltransferase subunits. Further, we identify the binding of a small protein, termed MtrI, to the cytosolic domain of MtrA, a previously unknown interactor occurring exclusively within, but not

ubiquitous across, the order Methanosarcinales. This binding is redox-dependent and occurs in response to cellular oxygen exposure, suggesting a potential role for MtrI in a protective mechanism against oxygen, supporting the adaptation of Methanosarcinales to microoxic environments in their natural habitats.

## Results

### The Mtr complex forms a distinct cloverleaf-shaped architecture

To achieve purification of the entire Mtr complex of *M. mazei*, a plasmid-borne variant of MtrE carrying a C-terminal TwinStrep-tag (TS-tag) was introduced into wild-type *M. mazei*, facilitating one-step affinity purification. This was achieved by employing an optimised version of the *Methanosarcina* shuttle plasmid harbouring the tagged MtrE subunit under the robust constitutive *mcrB* promoter.

Membrane fractions obtained from methanol-grown cells harvested during the exponential phase were solubilised using either n-dodecyl β-D-maltoside (DDM) or lauryl maltose neopentyl glycol (LMNG). Following solubilisation, Strep-Tactin affinity purification of the tagged MtrE successfully yielded a homogenous sample of the complete MtrA-H complex. The integrity and purity of the purified complex were analysed by SDS-PAGE, western blotting, size-exclusion chromatography, mass photometry, and tandem mass spectrometry analysis of tryptic peptides (Supplementary Fig. 1a–e). Mass photometry determined the molecular mass to be approximately 763 kDa, indicative of the full trimeric complex embedded within a detergent micelle (Supplementary Fig. 1d).

To unveil the molecular architecture of the Mtr complex, cryo-electron microscopy (cryo-EM) single-particle analysis was performed on the LMNG-solubilized sample. This analysis yielded an asymmetric reconstruction of the entire complex at a nominal resolution of 2.1 Å (Fig. 1A, B, Supplementary Figs. 2, 14 and Supplementary Table 1). The Mtr core complex consists of a trimer of hetero-nonameric protomers, each composed of three multi-spanning transmembrane subunits (MtrCDE), four single-spanning transmembrane subunits (MtrABFG), and a dimeric cytosolic methyltransferase, MtrH. The trimerization interface is formed by the single-spanning transmembrane subunits MtrABFG, creating a dodecameric core. Notably, these subunits possess an unusual length, extending twice the thickness of the membrane, forming a central stalk that protrudes towards the cytosol. This stalk interacts with three pseudo-symmetric heterotrimers formed by the multi-spanning transmembrane subunits (MtrCDE), resulting in a three-fold symmetric, cloverleaf-shaped architecture.

One of the stalk subunits, MtrA, features a cytosolic corrinoid-binding domain, MtrA$_{cyt}$, connected via a long, flexible linker. This domain harbours the 5-hydroxybenzimidazolylcobamide cofactor (Factor III, Supplementary Fig. 14), which serves as the mobile carrier element in the methyltransferase reaction (Fig. 1D and Supplementary Fig. 13). In our cryo-EM reconstruction, the flexible linker connecting this domain is not resolved, indicating high conformational mobility. Nevertheless, a subset of particles displayed a well-defined density corresponding to the MtrA$_{cyt}$ domain bound to the previously uncharacterised subunit MtrI, which in turn associates with the MtrCDE subcomplex. (Fig. 1A, B – side view). Notably, this configuration was observed only in a fraction of particles, giving rise to an overall asymmetry of the complex (Fig. 1A, B and Supplementary Fig. 2c). Consistent with this variability, regions encompassing MtrH and MtrA$_{cyt}$ displayed elevated flexibility, limiting the local resolution. To address this, we performed masked 3D classification, 3D variability analysis, and local refinements, yielding focused maps at 2.5 Å for MtrA$_{cyt}$ and 3.2 Å for MtrH (Supplementary Figs. 2c, 14). By combining these maps we created a composite map encompassing the full complex, which in turn allowed to construct a complete atomic model of the asymmetric Mtr complex (Fig. 1A, B and Supplementary Fig. 2c).

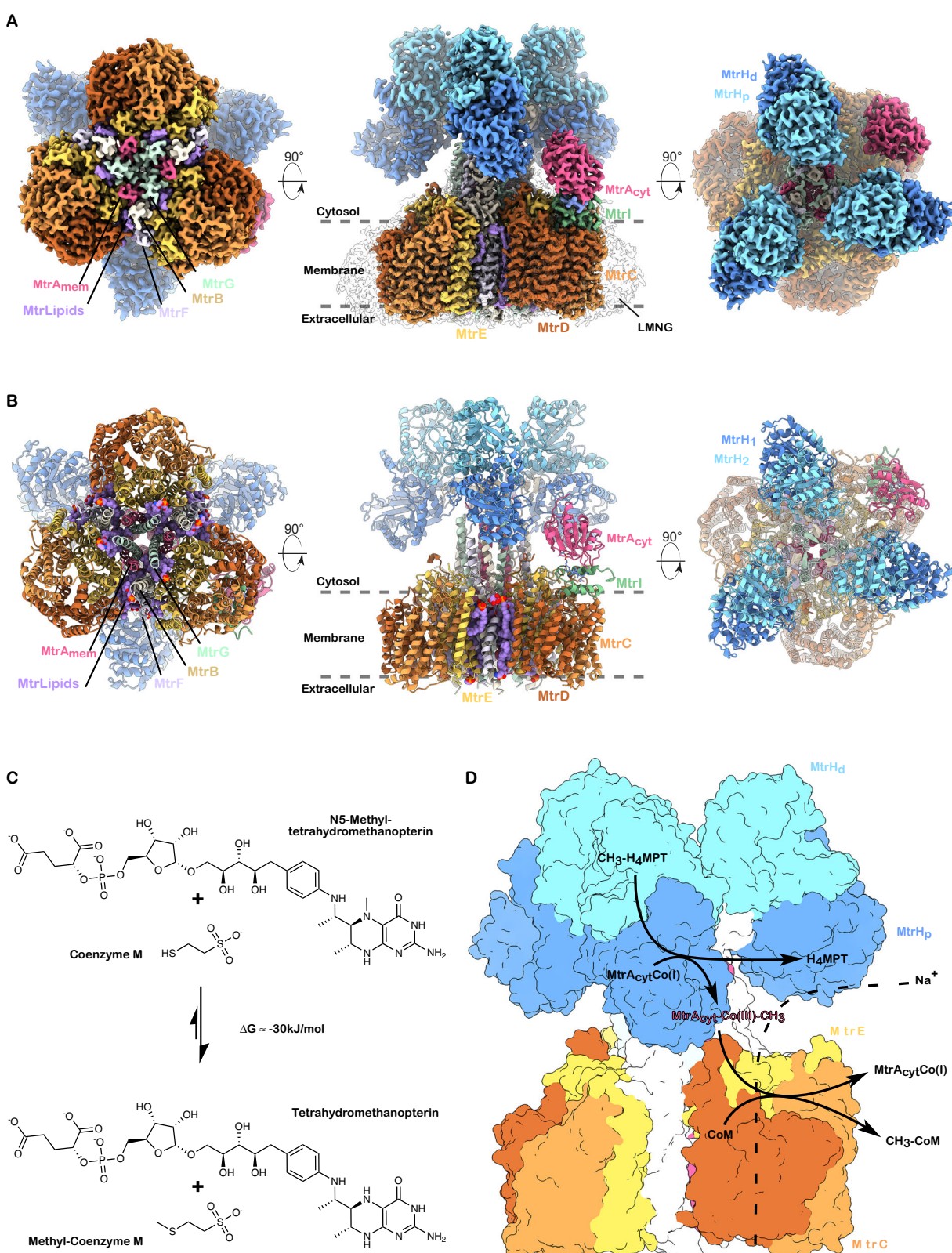

**Fig. 1 | Overall architecture of the *M. mazei* Mtr complex. A** Segmented cryo-EM composite map of the Mtr complex shown from different orientations. **B** Corresponding (atomic) model of the Mtr complex. In the side views of both map and model, membrane, cytosolic and extracellular compartments are labelled, and the position of the membrane is indicated by dashed lines. **C** Mtr-catalysed reaction with substrates and products shown as structural formulas. **D** Surface representation of the Mtr complex (with cytosolic MtrA and MtrI omitted). Curved arrows indicate substrate-to-product conversion, positioned at the site of catalysis within the surface model.

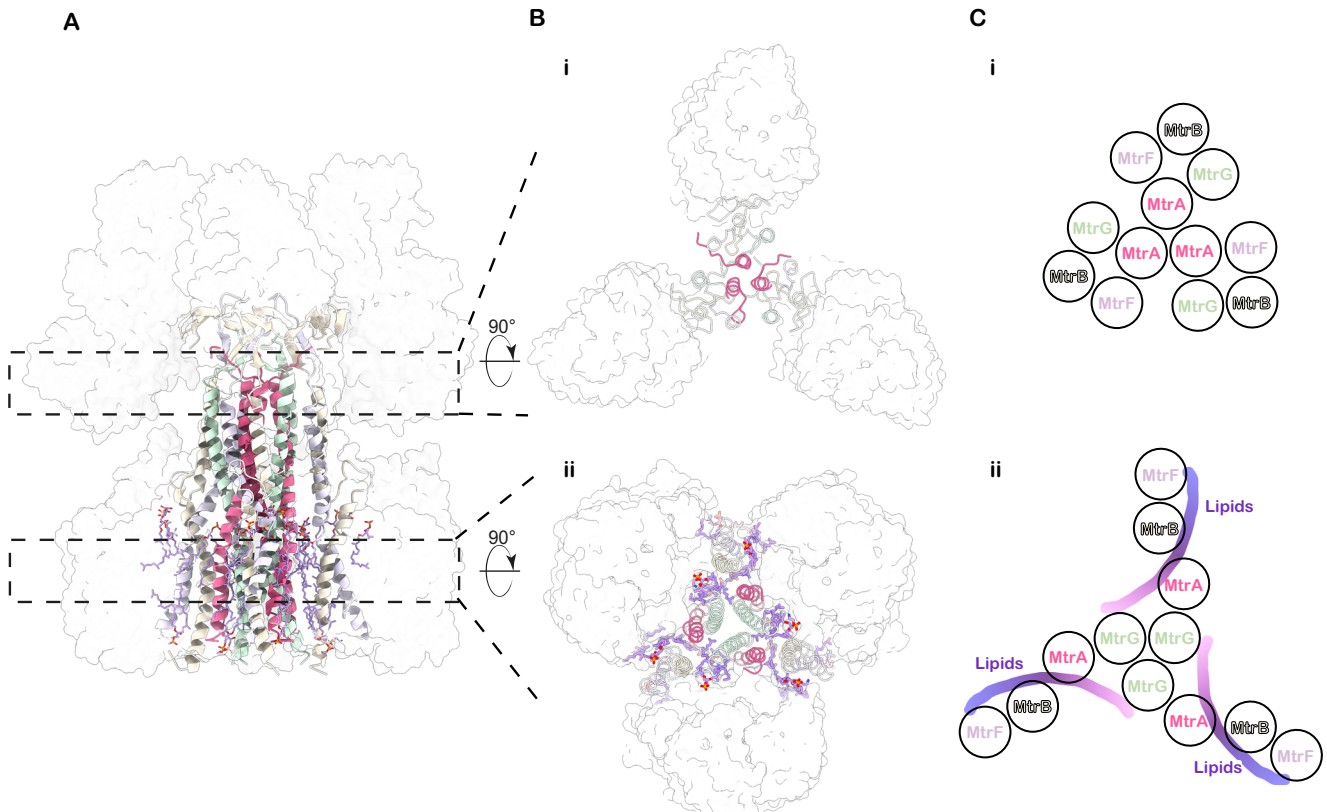

**Fig. 2 | Central stalk. A** Side view of the central stalk. **B i** Cross-section of the stalk in the cytosolic region, **(ii)** Cross-section of the stalk in the membrane region. **A**, **B** MtrH and MtrCDE subunits are shown as white, transparent surfaces. MtrBFG- and membrane-located MtrA-helices are shown as a cartoon. Archaeal phospholipids are shown as sticks. **C**, **i** Schematic representation of subunit arrangement in the cytosolic region, **(ii)** Schematic representation of subunit arrangement in the membrane region. The purple stripes mark the approximate position of the protein-embedded archaeal ether lipids (see also Supplementary Fig. 4).

## The central stalk forms a hydrophobic seal preventing ion leakage

The cytosolic section of the stalk (Fig. 2A(i), B(i), C(i)) features three straight helices from chain MtrA at its core, spanning residues A:177–210. Surrounding these, the helices from chains F, B, and G are arranged in a clockwise orientation (viewed from the membrane towards the cytosol) relative to chain A, aligning almost parallel to it. Hydrophobic interactions mediate this organisation, resulting in a tetrameric assembly (ABFG) on each side of the trimeric core, forming a trimer of tetramers (Fig. 2C(i)). This arrangement creates a vestibule connected to the cytosolic solvent, as indicated by resolved water molecules in our structure (Supplementary Fig. 3a). This vestibule is closed at the border to the membrane plane by three phenylalanines (Phe210) from MtrA (Supplementary Fig. 3b, c). In the membrane plane, the three MtrA helices bend outwards and are replaced as the core of the stalk by the MtrG helices (G44-G66), which form a three-helix bundle along the threefold axis. This bundle, characterised by conserved hydrophobic residues across methanogenic species, together with Phe210 from MtrA, acts as a tight hydrophobic seal that prevents ion leakage during conformational changes within the Mtr complex (Supplementary Fig. 3c).

Notably, helix B is discontinuous, featuring a cytosol-protruding loop spanning residues Pro56 to Thr70, which interacts directly with the membrane-spanning subunit MtrE. In addition, helices F and B wrap around and cross each other between residues MtrF:42–53 and MtrB:71–80, causing a pronounced tilt of MtrF away from the threefold axis. This disruption breaks the local tetrameric arrangement of MtrABFG within the transmembrane region. Within the resulting gap, a minimum of five well-defined archaeal ether lipids are bound per trimer (Fig. 2B(ii), 2C(ii) and Supplementary Fig. 4). These five lipids form an interacting lipid plane that weaves between the transmembrane helices of the membrane subunits. The density quality (Supplementary Fig. 14e) allowed us to model a total of 15 lipids, though additional, less ordered lipids may be present. These lipids are deeply embedded in the complex, acting as integral non-protein structural components of the Mtr complex, bridging and stabilising interactions between membrane-spanning subunits. Guided by our well-resolved cryo-EM map, the surrounding local environments, and considering the most abundant lipid species in *Methanosarcina*[25], we modelled these embedded lipids as either archaeol or 2-hydroxy-archaeol (2,3-di-O-phytanyl-sn-glycerol) bearing the polar headgroups phosphatidylethanolamine and monophosphate or phosphatidylinositol and phosphatidylserine.

## The dimeric methyltransferase MtrH

Like other related methyltransferase systems, MtrH exists as a homo-dimer of two MtrH monomers, each adopting a triose-phosphate isomerase (TIM) barrel fold. The structure reveals that three MtrH dimers are bound, resulting in a total of six MtrH copies within the complex. Similarly to its closest known structural relative, MtgA from *Desulfitobacterium hafniense*[26] the TIM-barrel is formed by eight parallel ß-strands in its centre, enclosed and connected by helices. The MtrH monomers dimerise via their C-terminal helices. The resulting MtrH dimer associates with the end of a three-stranded β-sheet formed by the N-terminal region of MtrB (residues 1–23; Fig. 3A, B). This ensures that MtrH consistently binds with the same orientation, positioning the TIM-barrel of the membrane-proximal MtrH (MtrH$_p$) to be opened in a clockwise way and the membrane-distal MtrH (MtrH$_d$) opened in a

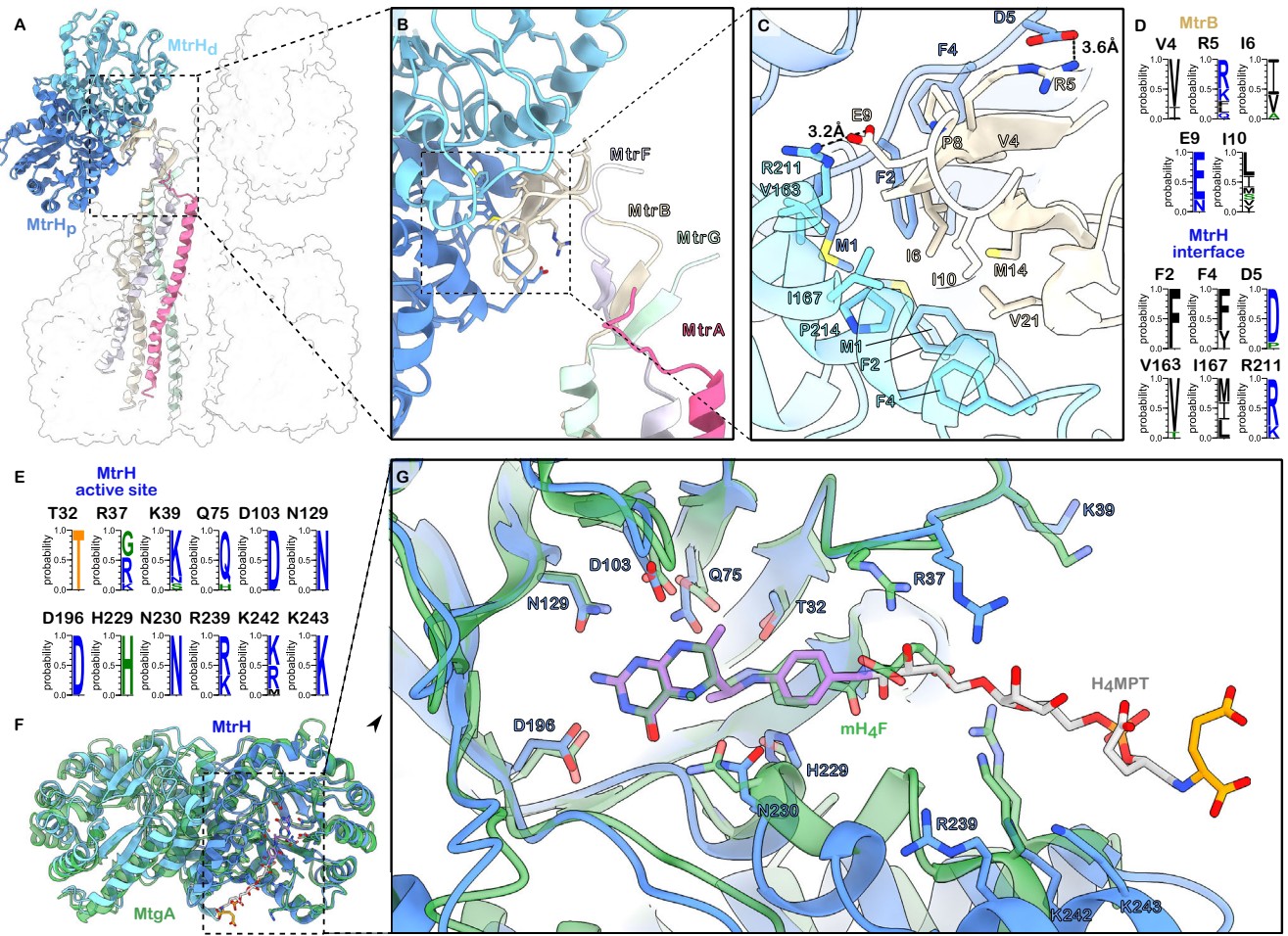

**Fig. 3 | Structure and interactions of the dimeric methyltransferase MtrH.**
**A** Overview of MtrH positioning in the Mtr complex. Cartoon representation of dimeric MtrH bound to one protomer of the stalk-forming MtrABFG helix-bundle. The rest of the complex is depicted as a white, semi-transparent surface. **B** Close-up view of the extended ß-sheet formed by the N-termini of MtrBFG and the MtrH-stalk interface created by the MtrB sheet. **C** MtrH-MtrB interface. Residues involved in hydrophobic and charged interactions are labelled and shown as sticks.
**D** Conservation of MtrB-MtrH interacting residues across Methanoarchaea.

**E** Conservation of substrate-binding residues within the MtrH active site across Methanoarchaea. **F** Superposition of MtrH (light and dark blue) and MtgA (PDB: 6SJN, green). **G** Close-up of the superposition of the active site of MtrH and MtgA. Residues involved or potentially involved in substrate-binding, as well as Methyl-tetrahydrofolate (mH$_4$F) and H$_4$MPT, are shown as sticks. The density of mH$_4$F (green) bound to MtgA was used to model the binding of H$_4$MPT (purple, grey, orange).

counter-clockwise way as viewed from the membrane towards the cytosol. Such an orientation facilitates the binding of the cytosolic domain of the methyl-group carrying MtrA. As the active sites of the MtrH dimer face in opposite directions, it is likely that the membrane-proximal MtrH$_p$ plays the primary role in binding MtrA. However, albeit unlikely due to steric restraints, interaction with the membrane-distal MtrH$_d$ cannot be fully excluded.

The interaction between the strands of MtrB and MtrH is mainly mediated by small hydrophobic residues that form a shape complementary to the hydrophobic binding pocket at the MtrH-dimerisation interface. In addition, it is stabilised via salt-bridges between Arg5$_{MtrB}$ and Asp5$_{MtrH}$ as well as Glu9$_{MtrB}$ and Arg211$_{MtrH}$ (Fig. 3C, D). This rather weak interface could be a reason for the often occurring loss of the methyltransferase in native purification procedures[19].

To better understand Tetrahydrosarcinapterin (H$_4$SPT) binding by MtrH, we superposed our MtrH structure with the structure of the homologous methyltransferase MtgA in complex with Methyltetrahydrofolate (mH$_4$F) (Badmann and Groll, 2020, PDB: 6SJN). Although the ligand is not present in the structure, the superposition reveals a highly conserved arrangement of residues within the pterin-binding site,

suggesting a conserved mode of pterin-moiety recognition. Residues Asp103, Asn129, Asp196, and Asn230 in MtrH are conserved and correspond to key residues in MtgA, suggesting they play identical roles in both methyltransferases. In addition, Thr32, Gln75, and His229, which also line the pterin-binding site, are not strictly conserved but preserve properties of their counterparts in MtgA and are therefore likely to fulfil analogous functions. This conservation of the binding environment enabled us to model H$_4$SPT into the substrate binding-pocket of MtrH by superposing the mH$_4$F ligand from MtgA. Further, comparative sequence analysis across methanogenic archaea identified an additional conserved feature: a positively charged patch formed by residues Arg239, Lys242, and Lys243 at the exit site of the enzymatic cavity. This conserved patch suggests a potential additional binding interface accommodating the polar and negatively charged ribosyl and gamma/alpha-glutamyl tail of H$_4$MPT/SPT. However, confirmation of the exact substrate-binding mode will require an experimental substrate-bound structure of MtrH.

**The CDE Trimer and the sodium ion binding site.** In the membrane region, the central MtrABFG stalk, together with several bound lipids, forms three equivalent binding sites for MtrCDE trimers. Each interface

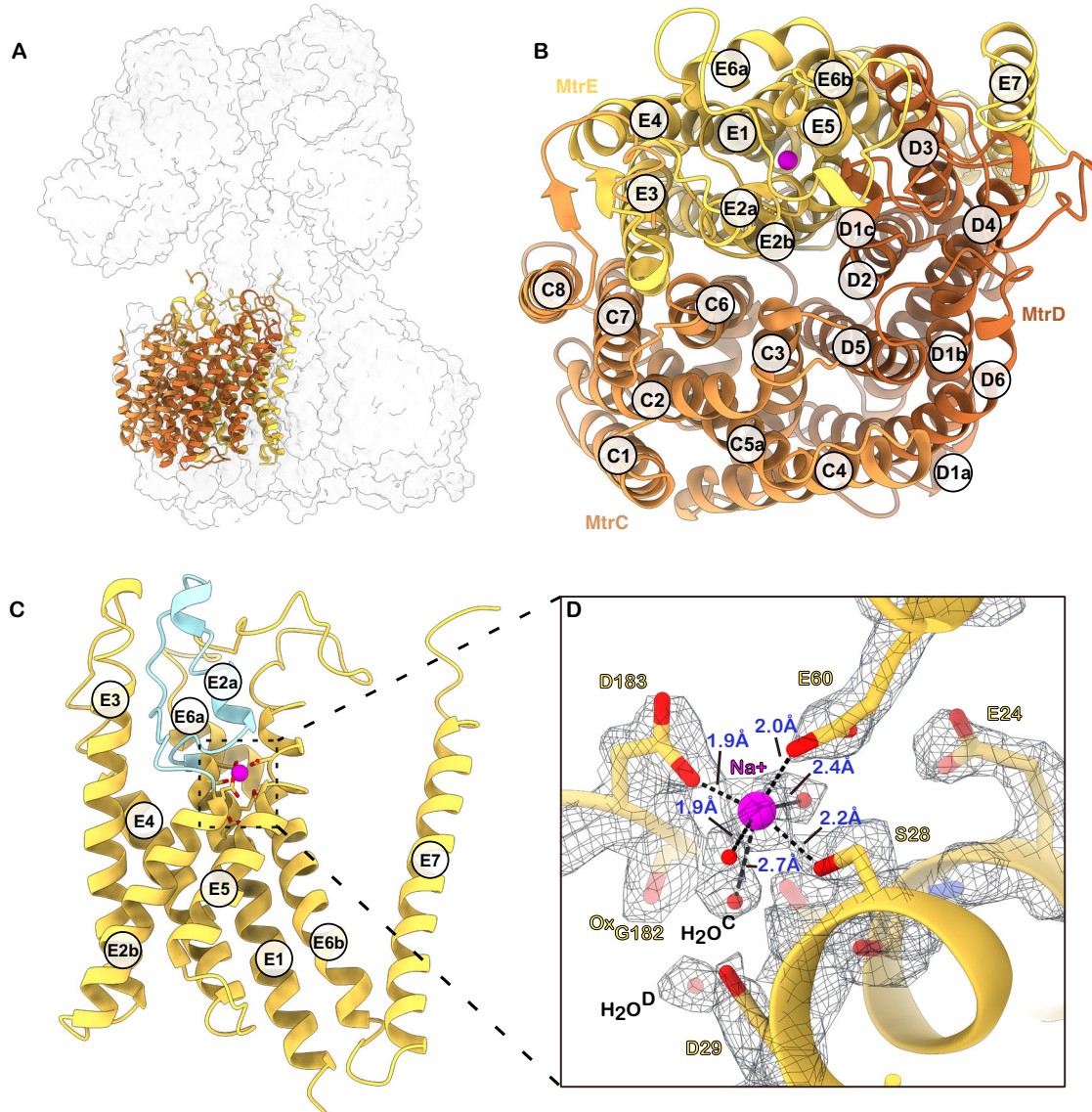

**Fig. 4 | The CDE membrane globe and sodium ion binding site. A** Overview of a single CDE trimer in the Mtr complex. **B** CDE trimer (cartoon) with helices labelled. Top-down view from the cytosolic side towards the membrane. The sodium ion is shown as a purple sphere. **C** Side view of MtrE (cartoon). Membrane-embedded loop-helix-loop structure highlighted in light blue. The sodium ion is shown as a purple sphere. **D** Close-up view of the sodium ion binding site within MtrE. Selected residues shown as sticks together with their cryo-EM density (threshold: 0.116). D183, E60 and S28 together with three water molecules form an octahedral coordination complex around a sodium ion (purple sphere). A previously proposed second sodium ion has been reassigned as a water $H_2O^D$.

is predominantly formed by MtrE, while MtrC and MtrD are positioned more peripherally toward the outer surface of the complex. The arrangement of one trimeric site can be followed in a clockwise direction when viewed from the membrane toward the cytosol (Supplementary Fig. 4b, d). Along this path, MtrE establishes extensive contacts with helices of MtrF and MtrB, archaeolphosphatidylethanolamine, one 2-hydroxy-archaeolphosphatidylserine, MtrG, MtrA, a second 2-hydroxy-archaeolphosphatidylserine, archaeolphosphate, and 2-hydroxy-archaeolphosphatidylinositol. Continuing clockwise, MtrC contributes to the interface through its transmembrane helices TM7 and TM8, which interact with archaeolphosphate and 2-hydroxy-archaeolphosphatidylinositol (Supplementary Fig. 4c–f). The C-terminal loop of MtrC projects toward MtrB and MtrG, making minor additional contacts with both.

MtrC, MtrD, and MtrE are each composed of six transmembrane helices organised into a pseudo-twofold symmetrical helical bundle (Fig. 4B). This internal symmetry allows TM1-3 to structurally superimpose onto TM4-6. Structural comparisons using DALI[27] reveal that these proteins possess a rare fold with no close structural analogues among known membrane proteins. However, modest similarities exist: MtrE shows resemblance to several translocating membrane proteins, including the sodium-pumping NqrE subunit of the NADH:ubiquinone oxidoreductase, the manganese importer PsaC, and the heme importer BhuU (Supplementary Fig. 5 and Supplementary Data 1). Despite low sequence identity, the resemblance to NqrE, which has evolved from RnfA[28,29] raises the possibility that this fold may have evolved to mediate sodium translocation under energy-limited conditions.

Interestingly, despite the shared overall architecture of MtrC, MtrD and MtrE, the sequence identity between these proteins is relatively low (MtrC–MtrE: 21.36%; MtrD–MtrE: 21.43%; MtrC–MtrD: 25%). This indicates these subunits likely descended from a common ancestor but diverged along distinct evolutionary paths, leading to specialised functions within the complex.

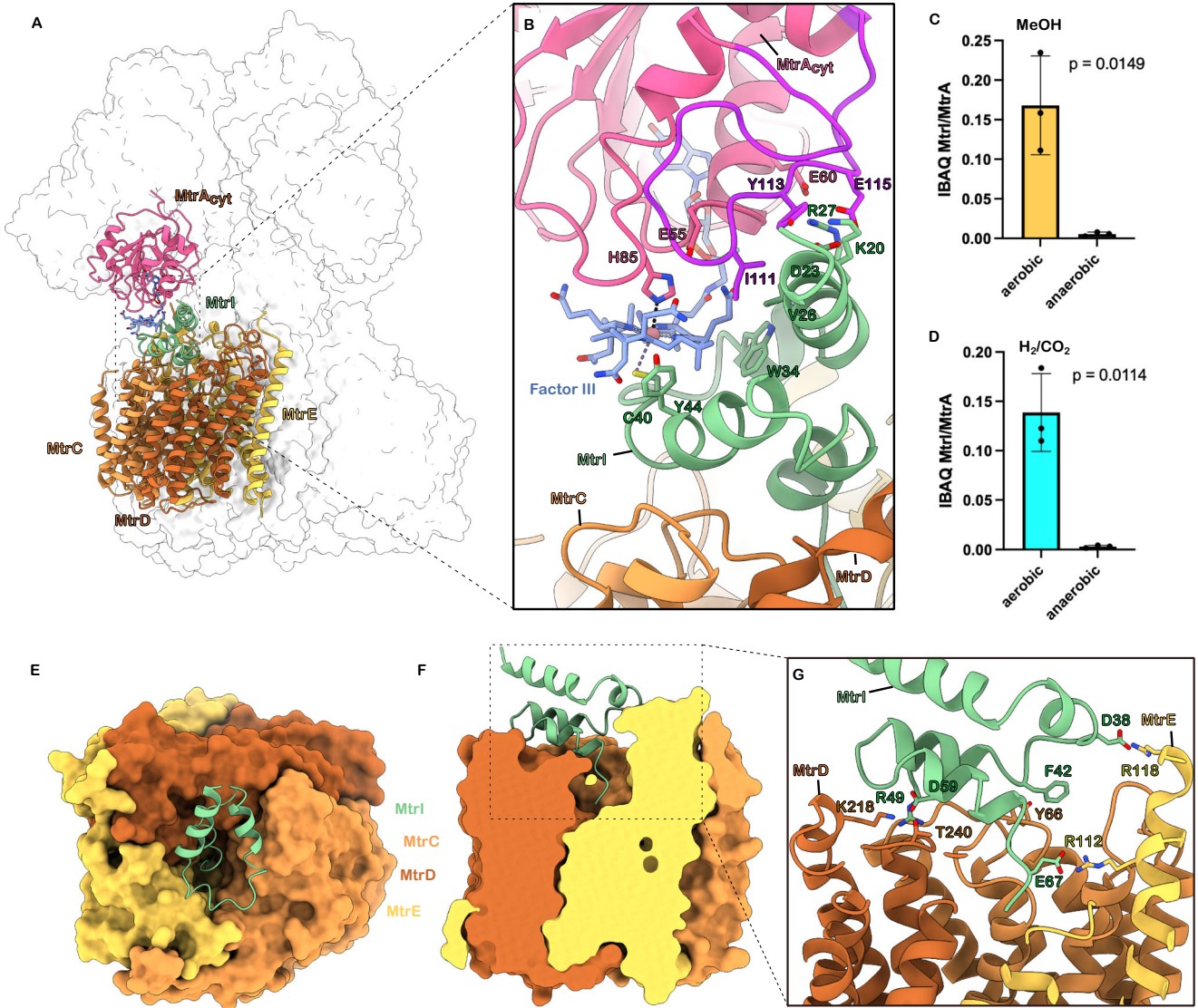

**Fig. 5 | MtrI binding to MtrCDE and MtrA is oxygen-dependent. A** Overview of MtrI (green) location in Mtr. MtrI is sandwiched by MtrA$_{cyt}$ (pink) and MtrCDE (orange, brown, yellow). **B** Close-up of MtrA·MtrI interaction. Interacting residues and Factor III shown as sticks and labelled. The unstructured insertion domain of MtrA is coloured purple. **C**, **D** Ratios of iBAQ values for MtrI and MtrA from Mtr complexes purified under oxic and anoxic conditions from methanol- (**C**) or H$_2$/CO$_2$-grown cells (**D**). Bars represent the mean ± SD of MtrI/MtrA ratios from $n = 3$ biological replicates. $P$-values were calculated from log$_2$-transformed ratios using a two-sided paired $t$-test. Methanol-grown cells (**C**): $t = 8.091$, df = 2, $P = 0.0149$, 95% CI of the mean difference = − 7.733 to − 2.364, partial $\eta^2 = 0.9704$. H$_2$/CO$_2$-grown cells (**D**): $t = 9.291$, df = 2, $P = 0.0114$, 95% CI of the mean difference = − 8.196 to − 3.008, partial $\eta^2 = 0.9774$. $P$-values are shown in the plot. **E** Top view of MtrI binding to the MtrCDE membrane globe (surface representation). **F** Side view of MtrI binding to the MtrCDE membrane globe (surface representation, clipped). **G** Close-up of MtrI·MtrCDE interaction. Interacting residues are shown as sticks and labelled.

Previously[17], sequence conservation analysis and the hallmark feature of high charge suggested that a signature loop in MtrE was a cytosolic loop containing a zinc-binding area. Contrary to this, our structural analysis reveals that this region in MtrE is actually a membrane-embedded element with a loop-helix-loop structure (Fig. 4C). The previously proposed highly conserved zinc-binding residues Asp25, Asp29, and Glu60, along with the suggested sodium ion-binding site residues Asp183 and Ser28, are in close proximity, forming a hydrophilic, charged pocket within MtrE. This pocket likely forms the sodium ion binding site of the Mtr complex. Our structure shows density consistent with a sodium ion octahedrally coordinated by six oxygen ligands with Na–O distances ranging from 1.9 to 2.7 Å (Fig. 4D), a geometry characteristic of Na$^+$ sites in protein structures[30]. Also, the second step of the Mtr-catalysed reaction has been demonstrated to be sodium dependent, with reaction rates increasing at

higher Na$^+$ concentrations and an estimated Kd for sodium binding of approximately 50 μM[11]. In addition, our samples contained 150 mM NaCl, ensuring saturation in binding under our experimental conditions. Further supporting this assignment, the coordinating residues are highly conserved among Mtr homologues (Fig. 4D). Finally, this interpretation is in agreement with the proposed sodium-binding site of a previously reported Mtr structure from *Methanothermobacter marburgensis*[20], PDB: 8Q3V, suggesting that this sodium-binding mode may be universally conserved across all methanogenic archaea.

## MtrI binds and connects MtrA$_{cyt}$ and MtrCDE
Unexpectedly, in our structure, the corrinoid-binding domain of MtrA does not directly interact with the integral membrane MtrCDE subcomplex. Instead, our cryo-EM map indicates that this interaction is bridged by a previously unidentified small helical protein (Fig. 5A). We

employed ModelAngelo (v.1.0)[31] to automatically build a model without a sequence input. Using *BLASTP* and *tBLASTn* with the output-sequence from ModelAngelo, the helical protein was identified as MM_2401, a small protein consisting of only 68 amino acids (Supplementary Fig. 6d). The respective gene is not part of the highly conserved *mtr* operon but represents a stand-alone gene and has so far evaded detection and attention. We named this protein MtrI.

MtrI consists of four helices (Fig. 5B and Supplementary Fig. 6e) that are lying atop the cavity of the MtrCDE subcomplex (Fig. 5E, F) stabilised by a series of charged and aromatic interactions (Fig. 5G): Asp38 interacts with Arg118 of MtrE, Phe42 forms aromatic interactions with Tyr66 of MtrC, Arg49 binds Lys224 of MtrD via a bridging water, and Asp59 interacts with Thr240 of MtrD. Crucially, the C-terminal residues of MtrI penetrate into the vestibule formed by the three MtrCDE subunits, where it is anchored by an interaction between the C-terminal Glu67 of MtrI and Arg112 of MtrE. This arrangement effectively blocks the central cavity of MtrCDE (Fig. 5F). Within the other two MtrCDE subcomplexes of the Mtr, we do not observe any MtrI. A stronger density next to Arg112 could be explained in accordance with another study[20] by a mixture of two partially occupied CoM sites. The presence of MtrI in the cavity, along with its interaction with Arg112, therefore potentially occludes the CoM binding site and blocks the sodium channel (Supplementary Fig. 8).

Superimposing MtrCDE not containing MtrI with the site of the trimer showing the MtrCDE–MtrI complex reveals no major structural differences. One notable exception is a slight rearrangement of the cytosolic loop in MtrC (residues 65–71), which includes Tyr66, to accommodate MtrI binding (Supplementary Fig. 7). This loop also appears more rigid in the cryo-EM density of the MtrCDE–MtrI complex.

Interestingly, MtrI features an N-terminal Zinc-ribbon domain comprising 15 residues, including 4 cysteines that form a zinc-binding motif (Supplementary Fig. 6e). However, in our cryo-EM map, the N-terminal region containing the Zinc-ribbon is highly flexible and poorly defined. Thus, we recombinantly expressed and purified N-terminally TwinStrep-SUMO tagged MtrI in *Escherichia coli* and demonstrated its Zinc-binding properties using ICP-MS (Supplementary Fig. 6f–i). Like most zinc-ribbons, this one likely serves a structural role, although its exact function remains unknown.

To further validate binding of MtrI to the Mtr complex, we performed a reciprocal pulldown experiment by expressing an N-terminal $His_6$-tagged variant of MtrI from a modified *M. mazei* shuttle plasmid in *M. mazei* (Supplementary Fig. 6b). Ni-NTA purification of these strains resulted in strong enrichment of Mtr subunits compared to the empty vector control (Supplementary Fig. 9a, b).

$MtrA_{cyt}$ of *M. mazei* shows a high degree of structural similarity to the previously reported structure from *Methanothermus fervidus* (RMSD = 0.599). As noted earlier, MtrA exhibits a distinctive binding mode of the corrinoid cofactor Factor III (FIII), setting it apart from other enzymatic systems[21]. A key feature of the interaction between MtrA and MtrI is the direct coordination of the cobalt atom of FIII by Cys40 of MtrI, which serves as the upper axial ligand (2.67 Å) and His85 of MtrA, which acts as the lower axial ligand (2.68 Å). Additional contacts are formed between Tyr44 and Trp34 of MtrI and the porphyrin ring of FIII (Fig. 5B). These interactions are further stabilised by a network of polar contacts involving Lys20, Asp23 and Arg27 of MtrI and the MtrA residues Tyr113 and Glu115 of the characteristic unstructured insertion domain of MtrA (residues 98–120), as well as Glu60.

**MtrI binding is redox-dependent and O$_2$-induced**. The unexpected presence of MtrI in our cryo-EM reconstruction prompted us to explore its potential functional relevance within the Mtr complex. Reanalysis of previously published ribosome profiling and mass spectrometry datasets from *Methanosarcina mazei* (PXD055748

[https://proteomecentral.proteomexchange.org/cgi/GetDataset?ID=PXD055748], PXD055745)[23] confirmed that MtrI is consistently translated and detected under all tested conditions, indicating a constitutive role in *M. mazei* physiology.

To assess its evolutionary conservation, we performed comprehensive sequence-based and structural homology searches. MtrI was found to occur exclusively, though not universally, within the order *Methanosarcinales*, specifically in the families *Methanocomedenaceae*, *Methanoperedenaceae*, and *Methanosarcinaceae*, including ANME-2a and −2b archaea (Supplementary Fig. 10 and Supplementary Data 2, 3). Despite moderate amino acid conservation, the genomic context of MtrI varies across genera (Supplementary Fig. 11), suggesting lineage-specific adaptation. Collectively, these findings indicate that *mtrI* is a conserved but non-essential accessory gene within Methanosarcinales, potentially conferring redox-responsive regulation of the Mtr complex.

When comparing Methanosarcinales with other methanogenic orders, two major differences stand out that hint at the function of MtrI: (i) Methanosarcinales can grow on additional substrates, facilitated by the presence of methanophenazines and additional energy-conserving systems that directly affect Mtr function, which, e.g., is reversed during methylotrophic growth (methanol, methylamines)[32], and (ii) Methanosarcinales often inhabit oxygen-rich environments[33–35]. To assess whether MtrI plays a function in mediating one of these physiological traits, we grew *M. Mazei* containing the MtrE-TS plasmid on either methanol or H$_2$/CO$_2$ and subsequently purified Mtr from these cultures under both oxic or strictly anoxic conditions.

To quantify MtrI abundance, we performed liquid chromatography-tandem mass spectrometry (LC-MS/MS) analysis of tryptic peptides derived from purified Mtr. Protein intensities and IBAQ values were calculated using MaxQuant (v.2.6.5.0)[36] and the MtrI:MtrA iBAQ ratio was used as a semi-quantitative metric to determine MtrI levels. For both growth conditions, H$_2$/CO$_2$ and methanol, oxygen-exposed purification of the Mtr complex consistently yielded similar levels of MtrI, which remained within the same order of magnitude (Fig. 5C, D). These oxygen-exposed purified samples appeared pink, indicating an oxidised Co(III) state. In contrast, Mtr samples purified under strictly anoxic conditions - yielding a brown-coloured sample that indicates a reduced Co(II) or Co(I) state - exhibited markedly lower levels of MtrI across both growth conditions (Fig. 5C, D). These findings suggest that MtrI association requires the oxidised Co(III) state of the corrinoid cofactor. Thus, MtrI binding to the Mtr complex appears to be redox-dependent and may be triggered within the cell upon exposure to oxygen.

## Discussion

Our findings advance the mechanistic understanding of the Mtr complex by providing a highly resolved cryo-EM structure, offering insights into its assembly and organisation. In contrast to previous suggestions of partial binding, our results reveal full occupancy of an MtrH dimer in the stalk region, which clarifies the dimer's role in the architecture of the complex. Furthermore, this study provides structural evidence of a small protein associated with a core bioenergetic complex in methanogenic archaea. This highlights the regulatory potential of small proteins, a class of the proteome that has been historically overlooked due to methodological constraints. This structure has also identified a single sodium-binding site, critical for understanding how sodium ion pumping is integrated with methyl transfer reactions in the energy conservation processes of methanogens. In addition, the structure implicates two possible regions were MtrA may alternatively and transiently bind - one at the membrane-proximal MtrH$_p$ TIM-barrel and another at the cavity formed by the MtrCDE trimer, where a potential CoM binding site has been proposed (Supplementary Fig. 13). While these associations are not directly resolved in the present cryo-EM

data, the inferred positions are consistent with the proposed trajectory of MtrA during methyl transfer[20]. Such a movement of MtrA could play a role in coupling the methyl transfer reaction with sodium ion pumping through conformational changes within the complex. However, despite these advancements, several critical gaps remain in our understanding of how these processes are coupled. To resolve this, further structural studies are needed under active turnover conditions.

A particularly exciting discovery was the identification of MtrI, a previously uncharacterised small protein encoded by an sORF, tightly associated with the Mtr complex in our preparation. This association was confirmed by reciprocal affinity purification of the Mtr complex using a His$_6$-tagged variant of MtrI expressed in *M. mazei*. Our structure further suggests that MtrI may recognise oxidatively modified or inactive forms of MtrA in the Mtr complex. It is known that the number of axial ligands that bind to cobalt in cobamide-cofactors is dependent on the redox-state of the cobalt centre. In general, Co(I) is ligand-free, Co(II) harbours one axial ligand (either upper or lower), and Co(III) binds two axial ligands (upper and or lower). The presence of Cys40 from MtrI as an upper-axial ligand and His85 of MtrA as a lower-axial ligand suggests a Co(III) oxidation state. This oxidation likely results from oxygen exposure during purification, suggesting that MtrI senses an oxidised cobamide cofactor. By blocking the MtrCDE cavity, MtrI might prevent sodium leakage and inhibit MtrA-CoM turnover. The stability of the Mtr complex under these conditions is uncertain. It could be marked for repair by connectase enzymes[37], activated reductively by as-yet uncharacterised ATP-dependent RaCo enzymes[38], or targeted for degradation. While the exact function of MtrI remains unclear, its association with Methanosarcinales' adaptation to oxygen exposure is particularly intriguing.

Unlike Class I methanogens, Methanosarcinales exhibit greater resilience to micro oxic conditions and inhabit diverse environments, such as wetlands, peat bogs, and oxygenated sandy sediments, where they frequently encounter oxygen exposure. The exclusive occurrence of MtrI in Methanosarcinales suggests that it evolved as an adaptation to oxygen-exposed environments. *Methanosarcina* are also known to employ mechanisms such as antioxidant enzymes, including catalase and superoxide reductase, as well as a high-affinity terminal oxidase, cytochrome bd (CydAB), to contend with oxygen and oxidative stress. Changes in thiol-molecule and polyphosphate (polyP) content, along with the development of biofilms, further enhance their ability to withstand oxidative conditions. Further studies are essential to clarify MtrI's precise role and determine whether it represents a key factor in oxygen resistance, potentially as part of a broader protective mechanism within Methanosarcinales. Besides our finding, there is further recent evidence that small proteins can modulate the activity of the Mtr complex[39].

Overall, this study has an impact that extends beyond our understanding of Mtr: it establishes the previously unrecognised role of small proteins in regulating core bioenergetic complexes in methanogenic archaea. By revealing MtrI to be a redox-responsive modulator of the Mtr complex, the study sets a precedent for small proteins as key regulatory elements in the methanogenic metabolism.

## Methods
### Chemicals
Unless specified otherwise, chemicals were acquired from Sigma-Aldrich, Carl Roth, Roche and Serva GmbH.

### Strains and plasmids
MtrE was expressed with a C-terminal TwinStrep tag (TS-tag) in *Methanosarcina mazei* DSM #3647 using plasmid pRS1743. pRS1743 was generated via Gibson Assembly and consists of the backbone pRS1595 (a shuttle vector for *M. mazei* and *Escherichia coli*)[40], the constitutive promotor *pmcr*B including a ribosome binding site (RBS) as a control element for the C-terminally TS-tagged *mtr*E gene

(*MM_1547*) (Supplementary Fig. 6a). In detail, the backbone pRS1595 was restricted with NotI and BamHI (New England Biolabs, Ipswich, USA) to obtain a linear fragment. Primers for the PCR-products (*pmcr*B, *mtr*E and TS-tag) were generated using NEBuilder (New England Biolabs, Ipswich, USA) and commercially synthesised by Eurofins Genomics (Ebersberg, Germany). The promotor *pmcr*B was PCR-amplified using the template pRS893 and primers Gibson_Promotor_for (5'agggccctaggtaccatatgGAGCTCTGTCCCTAAAAATTAAATTTTC3') and Gibson_Promotor_MtrE_rev (5'gtggttccatGTTTAATTTCCTCCT-TAATTTATTAAAATCAC3'); the *mtr*E -gene was amplified using chromosomal *M. mazei* DSM #3647 DNA as template and primers Gibson_MtrE_for (5'gaaattaaacATGGAACCACTCATAGGCATG3') and Gibson_MtrE_rev (5'accacgctgaTGCGGAAGCCTCCTCAGC3'); the TS-tag was PCR-amplified using template plasmid JS004 (a vector based on pET28a) and primers Gibson_twinstrep_MtrE_for (5'ggcttccgcaT-CAGCGTGGTCGCATCCC3') and Gibson_twinstrep_rev (5'actagtaacgt-taagcttgc**AAAAAAA**TTACTTTTCAAATTGAGGATGGGACC3'). The TS-tag reverse-primer contained additional A$_7$ that function as a transcriptional terminator in *M. mazei* (T$_7$ on mRNA level). The Gibson Assembly reaction was performed by using the Gibson Assembly® Master Mix (New England Biolabs, Ipswich, USA), 30 fmol restricted pRS1595, 90 fmol *mtr*E -PCR product and 210 fmol of *pmcr*B - and TS-tag-PCR-products each for 60 min at 50 °C. The assembly was then transformed into *E. coli* DH5α λpir following the method of Inoue[41]. The resulting plasmid pRS1743, which contained a C-terminally TS-tagged *mtr*E under control of the *pmcr*B promotor, was then transformed into *M. mazei\** cells and a single clone was isolated as described in Ehlers et al.[42].

His$_6$-MtrI was expressed in *M. mazei* using plasmid pRS2145 (Supplementary Fig. 6b). pRS2145 was generated by PCR-amplification of *His$_6$-MtrI* from template pRS2139[43] using primers His_MM_2401_N-deI_for (5'TTTCATATGCACCATCACCATCATCACATGC3') and His_MM_2401_NheI_rev (5'TTTGCTAGCTCAGTCCTCGACGTCAA-GAG3'). The PCR product and pRS1807 were restricted using NdeI and NheI (New England Biolabs, Ipswich, USA), ligated and transformed into *E. coli* Pir1 cells (Thermo Fisher, Waltham, USA). The resulting plasmid pRS2145 was transformed into *M. mazei\** cells by liposome-mediated transformation with sucrose according to Ehlers et al.[42].

MtrI (MM_2401) was expressed with an N-terminal TwinStrep-SUMO-tag (TS-SUMO) in *Escherichia coli* strain BL21 (DE3) from plasmid pEZ13. The plasmid was generated via Golden Gate Assembly from backbone JS004 (a vector based on pET28a). Coding sequence *mm_2401* was amplified from the *Methanosarcina mazei* DSM #3647 genome with forward (5'atgcaggtctcaatgaAATGTGAAGCATGTG3') and reverse (5'tacgtggtctcttcgaTCAGTCCTCGACGTC3') primers, and the SUMO coding sequence was amplified from plasmid pET28-His6-SUMO with forward (5'atgcaggtctcacatgTCGGACTCAGAAGTC3') and reverse (5'atgcaggtctcatcatTCCACCAATCTGTTCTCTG3') primers, containing matching directional overhangs produced upon BsaI cleavage. Primers for the expression of MtrI in *E. Coli* were commercially synthesised by Integrated DNA Technologies, Inc. (IDT, Leuven, Belgium). The plasmid was assembled using BsaI-HF®v2 and T4 DNA Ligase (NEB).

### Cultivation and harvest
*Methanosarcina mazei* Gö1 was grown at 37 °C, in closed, *anoxic* growth tubes in volumes ranging from small volumes (e.g., 5 mL in Hungate tubes) to larger volumes of up to 1 L in 2 L Duran bottles. The medium used was similar to DSMZ120 but buffered with 20 mM PIPES (pH 7.0) instead of NaHCO$_3$ and for cultures containing plasmids, puromycin was added at a final concentration of 2-5 mg/L (Alomone Labs). Media were prepared aerobically and turned anoxic after autoclaving by repeatedly cycling the gas phase with N$_2$. Complete anoxic conditions were achieved by the addition of 40 mg/L of Na$_2$S, followed by visually monitoring the reduction of resazurin. Vitamins, minerals

and antibiotics were added after autoclaving. Growth substrates used were either 250 mM methanol or an 80%/20% $H_2/CO_2$ atmosphere. When growing on $H_2/CO_2$, cultivation was performed in an orbital shaking incubator at 80-100 rpm. In general, growth was monitored by determining the optical density of the cultures at 600 nm (OD600). Cells were harvested during late log-phase at an OD600 of 0.7-1 at 10,000 x $g$ at 4 °C. For anoxic harvest, centrifugation bottles and lids were pre-incubated for at least 16 h under strict anoxic conditions inside of a vinyl anaerobic chamber with 95% $N_2$ and 5% $H_2$ (Coy Laboratory Products). Cultures were shuttled into the chamber and successively transferred into the centrifugation bottles. Tightly sealed bottles were moved outside the tent for centrifugation. Resazurin in the growth media served as an indicator to confirm maintenance of anoxic conditions throughout the process. Harvested cells were stored at − 80 °C.

### Purification of the Mtr complex via MtrE-TwinStrep Affinity Tag for subsequent Cryo-EM

Cells were suspended in lysis buffer [Buffer A (50 mM MOPS/NaOH pH 7.0, 10 mM $MgCl_2$, 150 mM NaCl) and 0.1 mg/ml DnaseI, cOmplete™ Protease Inhibitor Cocktail] in a volume ratio of 1 g cells per 10 ml of buffer. Cells were lysed by French press (1000 − 1200 psi) up to 3 times at 4 °C. The lysate was initially centrifuged at 20,000 x $g$ for 30 min at 4 °C to remove cell debris. For preparation of the membrane the supernatant was further centrifuged at 100,000 x $g$ for 1.5 h at 4 °C. The membrane pellet was solubilised in solubilisation buffer containing lysate buffer together with either 2.5 % w/v n-Dodecyl β-D-Maltosid (DDM) or 1.5 % w/v Lauryl Maltose Neopentyl Glycol (LMNG). The volume was chosen to yield a 1:2 w/w ratio detergent/membrane-pellet. Solubilisation was performed for 12-16 h at 4 °C on a slowly rotating roller mixer. Non-solubilised membrane was removed by a second round of ultracentrifugation (100,000 x $g$ for 1.5 h at 4 °C). The complex was further purified using a gravity column containing equilibrated Strep-Tactin Sepharose resin (IBA Lifesciences). The supernatant was incubated with Strep-Tactin Sepharose resin (IBA Lifesciences) for 1-2 h before its transfer to a gravity column. For the purification using LMNG, the resin was washed with Buffer A and the Mtr complex was eluted with Elution Buffer (Buffer A plus 2.5 mM desthiobiotin). For the purification using DDM, 0.05% w/v DDM was added to Buffer A and Elution Buffer. The protein complex was concentrated with a 100 kDa Amicon Ultra centrifugal filter (Merck Millipore). Protein concentration was determined by Bradford assay, using a calibration curve created with bovine serum albumin. Non-heated samples were analysed by 15% SDS-gels to determine protein purity and quality. Western blot with a horseradish peroxidase (HRP)-coupled anti-TwinStrep-tag antibody (IBA Lifesciences, cat. no. 2-1509-001) at a 1:5000 dilution was performed to visualise and confirm the presence of TS-tagged MtrE.

### TwinStrep-SUMO-MtrI expression in *E. coli*

For anaerobic protein production of TS-SUMO-MtrI, pEZ13 was transformed into *E. coli* BL21 (DE3) by the heat-shock method. Cells were cultivated in Lysogeny broth (LB). *E. coli* cultures were transferred into glass bottles upon reaching OD600 of 0.6–0.8, supplemented with 25 mM sodium fumarate, 0.5 % (w/v) glucose, 1 mM L-cysteine, 1 mM ferric ammonium citrate, 50 mM MOPS, pH 7.4, and 0.001 % (w/v) resazurin. Bottles were sealed with rubber stoppers, and the gas atmosphere exchanged with nitrogen. Protein expression was induced by the addition of 0.5 mM IPTG, followed by overnight cultivation at 20 °C. Protein was purified in an anaerobic chamber filled with 95 % $N_2$, 5 % $H_2$, with buffer anaerobized by successive degassing for 1 h, bubbling in an anaerobic chamber overnight, and addition of 2 mM DTT freshly before the purification. Cells were harvested by centrifugation and lysed using a Microfluidizer™ (Microfluidics™) in buffer (50 mM MOPS, pH 7.0, 10 mM MgCl2, 150 mM NaCl) with added lysozyme,

DNase, and 0.5 mM PMSF. Clear lysate was applied to Strep-Tactin® Sepharose (IBA Lifesciences), washed, and eluted with 2.5 mM Desthiobiotin (IBA Lifesciences). Aerobically, His6-Ulp1 was used to cleave off the TS-SUMO tag, and concomitantly removed by passing the eluate over HIS-Select® Nickel affinity gel (Merck). Protein was further purified by gel filtration using a Superdex™ 75 Increase 10/300 (cytiva), concentrated using a 3 kDa MWCO Amicon® (Merck Millipore), and analysed by Coomassie-stained SDS-PAGE. Metal content was analysed by ICP-MS.

### His6-MtrI expression and purification in *M. mazei*

His6-MtrI was expressed from plasmid pRS2145 in *M. mazei* and purified via His-tag-affinity chromatography-purification. The His6-MtrI-expressing cultures were grown in 1 L cultures on MeOH to an optical turbidity at 600 nm (OD600) of 0.65–0.85. From now on, each step was performed aerobically. Cells were harvested (6371 × $g$, 45 min, 4 °C), resuspended in 5 ml buffer A (50 mM MOPS/NaOH pH 7, 10 mM $MgCl_2$, 150 mM NaCl (Chemicals Carl Roth GmbH + Co. KG, Karlsruhe, Germany)) and cell disruption was performed twice using a French Pressure Cell at $4.135 × 10^6$ N/m$^2$ (Sim-Aminco Spectronic Instruments, Dallas, Texas) followed by centrifugation of the cell lysate for 30 min (6000 × $g$, 4 °C). The supernatant cell extract was solubilised using 1 % LMNG (Thermo Fisher, Waltham, USA) at 4 °C overnight, followed by a centrifugation step to remove insoluble debris (30 min, 6000 × $g$, 4 °C). His-tag-affinity chromatography purification was performed with a Ni-NTA agarose (Cube Biotech, Monheim, Germany) gravity flow column. The column-bound protein was washed with 20 and 50 mM imidazole (SERVA, Heidelberg, Deutschland) and the protein was eluted with 100 and 250 mM imidazole. The aerobic purification was also performed with *M. mazei* cells containing the empty plasmid pRS1595 as a control.

### Inductively coupled plasma mass spectrometry (ICP-MS) analysis

ICP-MS was used to measure metal concentrations. Briefly, acid digestion of 40 μl sample (0.09 mg/mL protein, as determined by Bradford assay) was performed in 11% (v/v) $HNO_3$ (Suprapur, Merck), for 3 h at 80 °C. This was followed by a 5.5 x dilution with Chelex-treated water, yielding a total volume of 250 μL per sample. Internal standards, indium (2 ppb) and germanium (20 ppb) (both from VWR Merck), were added to each sample for normalisation. Elemental analysis was conducted using an inductively coupled plasma mass spectrometer iCAP Q (ICP-MS, Thermo Fisher Scientific), equipped with an SC4DX autosampler (Elemental Scientific) and a MicroFlow PFA-100 nebuliser. Quantification of metals was done by comparison with serial dilutions of the ICP multi-element standard solution XVI (Merck). The ICP-MS operated with a reaction cell using a helium/hydrogen gas mixture (93/7%). Data acquisition was performed in triplicate using Qtegra software v2.18 (Thermo Fisher Scientific). Blank values and quality thresholds were calculated using protein-buffer standards. The measured concentrations (initially in ppb) were converted to molar units (μM) of metal per sample for quantitative analysis.

### Determination of substrate- and $O_2$-dependence of MtrI binding

Cells harbouring the MtrE-TS affinity tag plasmid were grown in triplicate in 2 L glass bottles either on Methanol (1 L total culture volume) or $H_2/CO_2$ (0.75 L total culture volume) to an OD600 of 0.6-1. Upon reaching the target OD, cultures were transferred to an anaerobic chamber (95% $N_2$ and 5% $H_2$). Inside the chamber, two culture bottles were carefully combined and split again into two equal halves. Successively, one half was taken outside for oxygen-exposed harvest, and the other half was filled into anoxic centrifuge bottles, sealed and then taken outside for anoxic harvest. Cells harvested under oxygen exposure were incubated in air for 1.5 to 2 h before flash-freezing in liquid

$N_2$. Cells were stored at $-80\,°C$ under oxic or anoxic conditions, respectively. To quantify MtrI bound to the Mtr complex in the presence or absence of oxygen, Mtr was purified from anoxically harvested cells under strictly anoxic conditions and from oxygen-exposed cells under oxic conditions. Mtr complex was purified as described above but in order to simplify the procedure and reduce the variations between anoxic and oxic purification a few steps in the protocol were changed. To lyse the cells, instead of a French press, a sonicator was used, with 3 times 3 min on/ 3 min off cycle at 50–60% power and a pulse-length of 0.5 s/s. Further, solubilisation was done directly with the cleared lysate, omitting both ultracentrifugation steps by adding LMNG to a final concentration of 1.5% (0.1 g LMNG per g wet cell weight) and for 12–16 h.

### Protein mass spectrometry, peptide identification and quantification using MaxQuant

For the mass spectrometry of purified Mtr complex 10 µl of concentrated Mtr complex was treated with 1% sodium lauroyl sarcosinate (SLS) and 1.5 µl of TCEP, heated for 10 min at 90 °C. Subsequently, 1.5 µl of 0.4 M iodacetamide was added for modification, followed by sample cleanup using sp3-beads. Samples were successively Trypsin digested overnight, desalted with C18 columns (Macherey Nagel, Chromabond C18 WP), dried in a Speedvac and reconstituted in 0.1 % TFA.

Peptides were analysed using liquid chromatography-mass spectrometry in an Orbitrap Exploris 480 (Thermo Fisher Scientific) coupled to an UltiMate 3000 RSLCnano system. For each sample 1–4 µl of the peptides were injected onto a C18 reverse-phase HPLC column using a 30-min gradient (0.15% formic acid to 0.15% formic acid with 35% acetonitrile). The quantity of peptides injected was consistent across different samples within each experiment: 0.06 µg for $H_2/CO_2$ samples, 0.1 µg for methanol samples and 0.1 µg for $His_6$-MtrI-pull-down samples. The mass spectrometry data were analysed initially using Proteome Discoverer 1.4 (Thermo Fisher Scientific) against the proteome of *Methanosarcina mazei* (DSM #3647). Further, the raw MS data were analysed using MaxQuant (v.2.6.5.0). Protein search and identification was done using the integrated Andromeda search engine against the *Methanosarcina mazei* (DSM #3647) proteome. Only trypsin/P-specificity was considered, and up to two missed cleavages, a minimal length of 7 residues, fixed Carbamidomethylation and variable methionine oxidation and N-terminal protein acetylation. Orbitrap default settings were applied with a first search precursor tolerance of 20 ppm, a main search precursor tolerance of 4.5 ppm and a fragment ion tolerance of 20 ppm. A maximum false detection rate (FDR) of 1% was used using a target-decoy approach. Calculation of Label-free quantification (LFQ) and intensity based total quantification (iBAQ) were enabled. The remaining parameters were kept as default settings. To semi-quantitatively determine MtrI abundance, iBAQ values for MtrI and MtrA where divided and plotted as bar-plots. *P*-values were calculated using a paired t-test (two-tailed) in GraphPad Prism (v.10) of $log_2$-transformed iBAQ values. For $His_6$-MtrI pulldowns, iBAQ values of biological triplicates ($His_6$-MtrI pulldowns) and biological duplicates (vector control) were ranked according to their mean values and plotted against their rank. Plots were made in GraphPad Prism (v.10).

### Molecular size determination

We performed size exclusion chromatography (SEC) by injecting 400 µl of DDM-solubilised, purified protein into a Superose 6 Increase 10/300 GL column (Cytiva), previously equilibrated with buffer A and 0.05% DDM and attached to an Äkta pure system. Chromatography was performed at 4 °C and a flow rate of 0.4 ml/min. Protein elution was monitored by UV-absorbance at 280 nm. No size-exclusion chromatography was performed with the LMNG-solubilised sample prior to Cryo-EM sample preparation, as the StrepTactin purification already yielded a highly pure preparation. LMNG is known to bind membrane

proteins strongly and can be used at sub-CMC concentrations, which prevents the formation of empty detergent micelles. Therefore, a further SEC step after affinity purification was not required. We further confirmed the integrity, homogeneity, and stability of the complex for cryo-EM analysis using mass photometry (Supplementary Fig. 1c, d). To analyse the molecular weight of the protein complex by mass photometry a OneMP mass photometer was used and data was acquired with AcquireMP v.2.3 (Refeyn). As mass photometry of Mtr was impossible in the presence of detergent-micelles as in the DDM-solubilised sample, we only used Mtr complex that was solubilised with LMNG – but washed and eluted without detergent. Movies were recorded at 1 kHz, with exposure times ranging from 0.6 to 0.9 ms, to optimise signal while avoiding saturation. Glass slides were cleaned with isopropanol, and Mili-Q water and silicon gaskets were placed onto the clean slides. Following instrument calibration, 19 µl of buffer A was pipetted into a gasket well before finding the focus. The focal position was locked using the autofocus function of the instrument. The measurement started after adding and mixing 1 µl of 0.5 µM protein onto the gasket well. Data analysis was performed using DiscoverMP software (Refeyn).

### Cryo-EM and grid preparation

Grids for cryo-EM of the Mtr complex were prepared using QUANTI-FOIL R 2/1 copper 300 mesh grids (Quantifoil Micro Tools). Prior to sample application, the grids were glow-discharged for 25 s at 15 mA using a PELCO easiGlow device (Ted Pella). LMNG-solubilised Mtr at a concentration of 6 mg/mL was pre-mixed in a 9:1 ratio with 10 mM CHAPSO, resulting in a final CHAPSO concentration of 1 mM, to counter preferred particle orientation. Immediately after mixing, 4 µL of the protein solution was applied to the grid and plunge-frozen in liquid ethane using a Vitrobot Mark IV (Thermo Fisher Scientific). Grids were blotted for 6 seconds with a blot force of 6, while the chamber-atmosphere being kept at 4 °C and 100% humidity.

### Data collection and processing

Cryo-EM data for LMNG-solubilised Mtr was acquired using Smart EPU Software on a TFS Krios G4 cryo-TEM operating at an accelerating voltage of 300 keV and equipped with a Falcon 4i Direct Electron Detector (Thermo Fisher Scientific) and Selectrics Energy Filter set at a 10 eV slit width. Data were acquired at a nominal magnification of 165,000 x corresponding to a calibrated pixel size of 0.73 Å. Images were acquired with an exposure dose of 60 e⁻ Å⁻² in counting mode and exported as an electron-event representation (EER) file format. A dataset containing 21,552 micrographs was acquired. The entire processing was done in CryoSPARC[44] v.4. The EER files were fractionated into 80 frames, motion corrected using patch motion correction[45], followed by contrast transfer function (CTF) estimation. Importing beam-shifts and successive optical grouping (38 groups) improved the later estimation of higher-order aberrations. Manually picking of 500–600 particles was followed by training of a Topaz model[46]. Topaz extract of all micrographs led to an initial particle set of 1.1 million particles. Particles were initially extracted at a box size of 450 pixels and subjected to heterogeneous refinement to sort particles in 3D. An initial map, that was generated from a testing dataset, was used three times as an input seed. One of the three heterogeneous refinement output maps, containing 450,704 particles, was further re-extracted at a box size of 576 pixels and refined to 2.44 Å. Masked 3D classification without alignment (20 classes, filtered at 6 Å) was performed by placing a soft mask on the MtrA subunit. Masks were created with the ChimeraX map eraser tool or the molmap command, followed by map filtering and successive CryoSPARC import. Six out of 20 classes (139,221 particles) showed MtrA bound to the membrane plane. The particles were subjected to non-uniform refinement followed by reference-based motion correction and another non-uniform refinement yielding the Mtr consensus-map at a resolution of 2.06 Å with $C_1$-symmetry. To better resolve MtrA, we applied local refinement using

the same MtrA-mask as in the 3D-classification, yielding a local map at a resolution of 2.49 Å. To resolve MtrH, we created a soft mask around the flexible MtrH dimer region and performed 3D variability analysis with five modes to solve at a filtered resolution of 5 Å. Successive 3D Variability Display of a single component in cluster mode led to 20 clusters filtered at 5 Å, seven of which showed MtrH in similar positions. These seven clusters, containing 225,093 particles, were further subjected to masked 3D classification using the same MtrH mask. Out of four classes, one (56,229 particles) was locally refined to obtain a map of 3.2 Å resolution. To create a complete Mtr composite map, MtrH was further $C_3$-symmetrized using Volume alignment tools with 120° 3D rotations. The composite map was constructed in ChimeraX using the vop max command upon normalising the maps to the same threshold with vop scale. The final composite incorporated the consensus map, the locally refined MtrA and the $C_3$-symmetrized MtrH.

### Model building and refinement

Initial models were built with AlphaFold2[47]. To interpret the unknown density between MtrCDE and MtrA, we used ModelAngelo (v.1.0)[31] without providing a sequence and using the consensus map as an input. ModelAngelo modelled a continuous peptide of 52 residues into the density. A protein BLAST showed the highest similarity to a hypothetical protein that is conserved in *Methanosarcina* species. As the protein was not annotated in the protein database, we performed *tBLASTn* to retrieve the corresponding *M. mazei* sequence. The best-matching result by far was a reading frame of 204 nucleotides encoding a 68-residue protein with the locus name MM_2401 and the UniProt accession Q8PUD4. ChimeraX[48] and Coot (v.0.9.8.91)[49] were used for manual model building. The presence of water molecules was predicted in the consensus map using douse, as implemented in the PHENIX package (v.1.21-5207), with standard settings. Real-space refinements of models were performed iteratively with PHENIX. Ramachandran, reference model and secondary structure restraints were applied during refinement. CIF files for ligands like the etherlipids were created using AceDRG[50] in the CCP4 suite or eLBOW in PHENIX. Maps were graphically depicted using ChimeraX and Coot.

### Sequence conservation analysis

Sequences of MtrH and MtrB of *Methanococcus maripaludis* S2 (MtrH: CAF31123.1, MtrB: WP_181487237.1), *Methanocaldococcus jannaschii* DSM 2661 (MtrH: AAB98859.1, MtrB: WP_010870364.1), *Methanothermobacter marburgensis* str. Marburg (MtrH: CAA59003.1, MtrB: WP_237779309.1), *Methanobrevibacter smithii* ATCC 35061 (MtrH: ABQ87212.1, MtrB: WP_004032922.1), *Methanopyrus kandleri* AV19 (MtrH: CAA74773.1, MtrB: WP_236953805.1), *Methanothrix soehngenii* GP-6 (MtrH: AEB67809.1, MtrB: MDD3552885.1), *Candidatus Methanoperedens nitroreducens* (MtrH: KPQ42840.1 MtrB: WP_096206332.1), *Methermicoccus shengliensis* (MtrH: HIH70342.1, MtrB: WP_042687030.1), *Methanosarcina mazei* Go1 (MtrH: AAC38337.1, MtrB: AAC38333.1), *Methanosarcina acetivorans* C2A (MtrH: AAM03722.1, MtrB: AAM03726.1), were aligned using Clustal Omega[51]. Sequence logos were generated with WebLogo 3.0[52].

To test for the MtrI conservation, all of the available euryarchaeota genomes (7873 genomes) were downloaded (2nd of July 2024) using NCBI datasets command line tools v.16.22.1 (https://www.ncbi.nlm.nih.gov/datasets/docs/v2/command-line-tools/download-and-install/). We predicted the quality of all euryarchaeota genomes (isolates and metagenomes assembled genomes (MAGs)) using CheckM2 v.1.0.2[53] and the taxonomy using GTDB-Tk v.2.4.0 together with database release R226[54,55] Subsequently, we used tblastn v2.13.0 +[56] with the aa-sequence of MtrI (MM_2401) as query against all downloaded euryarchaeota genome nucleotide sequences. Applying cut-offs at 50 % completeness and <10 % contamination resulted in

7685 genomes, with 1216 belonging to the class of Methanosarcinia, matching to 271 species. MtrI was considered present in a species, when at least one genome of the respective species contained MtrI with ≤ 1 × 10⁻⁵ e-value, ≥ 70 % coverage and ≥ 30 % sequence identity. Furthermore, web-based searches with the aa- and nucleotide sequence, as well as with the experimental structure of MtrI, were performed. Web-based blastn, blastp, and tblastn (tools by NCBI[56,57]) were performed in order to blast against the currently available genome collections (accessed 21st of August 2025, databases used were nr/nt, nr and nt, respectively). Furthermore, Foldseek (accessed 17th of September 2025, E-value ≤ 1 × 10⁻³, > 30 % sequence identity) and DALI (accessed on 25th of September 2025) was used in order to check for structural conservation[27,58].

### Reporting summary

Further information on research design is available in the Nature Portfolio Reporting Summary linked to this article.

### Data availability

The cryo-EM maps generated in this study have been deposited in the Electron Microscopy Data Bank (EMDB) under the following accession codes: EMD-53361 (Composite map), EMD-53359 (Consensus map), EMD-53360 (MtrA local refinement), EMD-53358 (MtrH local refinement). Corresponding coordinate files have been deposited in the RCSB Protein Data Bank (PDB) under the following accession codes: 9QTS, 9QTQ, 9QTR, 9QTP. Coordinate files for MtgA are accessible under the accession code 6SJN. Coordinate files for Mtr (*M. marburgensis*) are accessible under the accession code 8Q3V. The mass spectrometry proteomics data generated in this study have been deposited to the ProteomeXchange Consortium (http://proteomecentral.proteomexchange.org) via the PRIDE partner repository[59] with the dataset identifier PXD064689. Previously published, cited *Methanosarcina mazei* proteomics data can be accessed under the identifiers PXD055748 and PXD055745. The DALI and Foldseek search results generated in this study are provided in the Supplementary Information and as Supplementary Data files. Source data are provided as a Source Data file. Source data are provided in this paper.

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

## Acknowledgements

We acknowledge the contributions of the cryo-EM Facility of Philipps-University Marburg. We acknowledge the cryo-EM Platform (CEMP), Helmholtz Munich, for data acquisition. We thank Rolf Thauer for continuous support and helpful discussions. We thank Darja Deobald from the UFZ Leipzig for performing metal quantification through ICP–MS. We thank Georg Hochberg from the University of Marburg for enabling access to mass photometry. We thank Cynthia Chibani from Kiel University for running the conservation analysis pipelines and providing valuable input and support concerning the output interpretation. This work was supported by the Deutsche Forschungsgemeinschaft Emmy Noether grant (SCHU 3364/1-1, to JMS), RTG 2937 (project number 505997786) to JMS and SCHM1052/20-2 to RAS, as well as the European Union's Horizon 2020 research and innovation programme (Two-CO$_2$-One; grant agreement no. 101075992) to JMS. The views and opinions expressed are those of the author(s) only and do not necessarily reflect those of the European Union or the European Research Council. Neither the European Union nor the granting authority can be held responsible for them. T.R.-T. acknowledges the funding from the International Max Planck Research School, Principles of Microbial Life.

## Author contributions

**E.H**. cultivated cells, constructed plasmids, transformed cells, screened mutant strains and performed the genome conservation analysis. **T.R.T**. cultivated cells, isolated and characterised the complex, vitrified cryo-EM grids and performed the mass spectrometry data analysis. **J.K**. carried out the mass spectrometry measurements. **E.Z**. constructed plasmids and purified proteins. **A.K**. and **S.B**. performed the initial grid screening and cryo-EM data collection. **T.R.T., T.P**., and **A.K**. conducted the cryo-EM data analysis, model building, and refinement. **T.R.T., T.P., A.K**., and **J.M.S**. interpreted the cryo-EM structures. **T.R.T**., **E.H**., **J.M.S**., and **R.S**. wrote the manuscript from the input of all other authors.

## Funding

## Competing interests

The authors declare no competing interests.
