## [Transparent Peer Review file · Nature Communications]

Structure of the *Methanosarcina mazei* Mtr complex bound to the oxygen-stress responsive small protein MtrI

Corresponding Author: Dr Jan Schuller

Version 0:

Reviewer comments:

Reviewer #1

(Remarks to the Author)

Tristan Reif-Trauttmsdorff et al. describe in their manuscript the cryoEM structure of the methanogenic Na⁺ translocating methyl transferase complex Mtr. They report on the overall structure and possible catalytic mechanism including a previously unrecognized Na⁺ binding site. In the course of their characterization, they discover a small regulatory protein MtrI present only in Methanosarcinales, possibly aiding against oxidative stress.

Overall, the authors report novel findings on an important enzyme of the methanogenesis pathway. They build on work done previously, both in sequence-based and structure-based investigations and add new pieces to the puzzle.

I have two main parts I'm critical about. The first is the way that the authors express their statements. They are often very bold and not refined to appropriately reflect the more nuanced state of the art. For non-experts reading this paper this can lead to severe misunderstandings of the field so I urge the authors to correct this. In my detailed comments I've named them where they appear. The introduction is also scarce in use of literature.

The second critique is directed at their analyses and statements surrounding the phylogenetic distribution of mtrI. The authors neither report how they determine that mtrI is restricted to Methanosarcinales (sequence or structure based search? Both need to be done. Cutoffs? Methods?) nor do they show any data that all Methanosarcinales carry mtrI. This analysis must go beyond a quick BLAST search but rather encompass comparisons of genome content on a larger scale. If orfs are not appropriately annotated for small such small proteins they may be missed by simple database searches, leading to faulty reporting; primary data genome comparisons may be needed. In order to satisfy this criticism, new and extensive in silico analysis needs to be performed currently not reported on in the manuscript.

Line numbers would have been helpful to facilitate the review process.

Abstract first sentence: contribute reads strange, emit?

Introduction

Anaerobic conditions should be anoxic conditions. Metabolism is anaerobic, conditions are anoxic.

The represent the only organisms... while I understand the notion of the author's, there are numerous other metabolisms that produce methane in trace amounts, e.g. demethylation reactions of methylphosphonates. The sentence as is does not represent the current state of the field. Please rephrase the sentence carefully.

Why is 'greenhouse gas' in brackets? I would replace climate relevant gas with greenhouse gas

There are several, phylogenetically not related pathways: I disagree with this sentence. Of course they are phylogenetically related, methanogenesis was an ancestral trait of the last common ancestor of methanogens. I urge the authors to phrase their statements more carefully.

This sodium-motive force drives chemiosmotic energy conservation and ATP synthesis: please add appropriate citations for this statement

Recent advances in ribosome profiling...: it would be nice to see some literature references underpinning these statements.

Results

Methanosarcina in italics

McrB promotor should be mcrB promotor (mcrB in italics)

Of a trimer of a: remove a

Highlighting the conservation of this binding sites in all methanogens: by comparing two structures one cannot conclude that this is conserved in all methanogens. Please phrase more carefully.

MtrI apparently coordinates the cobalt atom of FIII; this seems like an essential function to me while MtrI is constrained to Methanosarcinales. In my opinion the authors need to mention how they think non-Methanosarcinales Mtr coordinates the cobalt atom of FIII if MtrI is absent

Discussion

...with a core bioenergetic complex in Methanoarchaea: no, not in Methanoarchaea. In Methanosarcinales.

I think the authors very quickly conclude that MtrI is absent in methanogens outside of Methanosarcinales. I couldn't find in their methods how they checked. Only sequence based BLAST analysis (threshold for identity?), or also structure-based similarity searches? Both are required, and thresholds must be provided, defining presence/absence of homologous proteins. As the authors included ANMEs in their introduction I also expect to read whether ANME harbor mtrI.

The authors state that Methanosarcinales harbor mtrI. Have they checked that all Methanosarcinales contain them? This is not a quick analysis but I think is required if the authors would like to state that all Methanosarcinales have them. The way how they check this needs to be described, too.

I was also disappointed that the authors did not take available datasets on transcriptomes and proteomes to search for expression/presence of MtrI in those datasets.

Materials:

How were E. coli buffers made anoxic? By addition reducing agents? Please add to both E. Coli paragraphs
How much sample was used for ICP-MS?

Reviewer #2

(Remarks to the Author)

Summary

The manuscript describes the structure of the Mtr complex, a methyl-transferase from methanogenic archaea. Based on the structure, the authors identify a novel complex component and propose an oxygen-stress response mechanism. The results are relevant to bioenergetics, structural biology and from a broad evolutionary perspective and the data is solid. However a significant amount of the described results is not shown in figures. This makes comprehension of the manuscript difficult. I therefore recommend that the authors expand the manuscript by including the missing representations, in addition to addressing minor points listed below.

In essence, the results are noteworthy, novel and of significance for a broad audience across multiple disciplines and the methodology is sound. However, for the sake of clarity and reproducibility all the results that are described and discussed should be shown and this is currently missing in some parts of the manuscript. I therefore recommend the article is revised and updated accordingly.

Main comments

-For resubmission, line and page numbers should be added.

-On page 2, the meaning of the sentence "In methylotrophic methanogens that disproportionate methanol into methane and CO₂, this reaction is reversed" is not clear. Particularly, it is not clear what "disproportionate" means in this sentence.

-Also on the same page, the sentence "anaerobic methanotrophic archaea (ANME) oxidize methane to CO₂ operate the methanogenic pathway and thus the Mtr enzyme entirely in reverse." is unclear. Maybe punctuation is missing?

-Fig S1c should include the SEC profile of the LMNG-solubilised sample, since this is the sample used for structure determination, instead of the DDM-solubilised material.

-In Fig 1, indicate (1) the membrane in the side views of the map and model, as well as (2) both of the soluble compartments on the two sides of the membrane.

-Also in Fig 1 add a panel highlighting the position of the active sites and indication of the reactions (substrates and products), to clarify which reactions happen where in the complex.

-On page 5, the paragraph

"One of the stalk subunits, MtrA, features a cytoplasmic corrinoid-binding domain, MtrAcyt, connected via a long, flexible linker. This domain harbours the 5-hydroxybenzimidazolylcobamide cofactor (Factor III), that serves as mobile carrier element in the methyltransferase reaction. Our cryo-EM analysis revealed that only a subset of particles exhibited a well-resolved MtrAcyt domain bound to a single MtrCDE subcomplex, resulting in an overall asymmetry of the complex. Furthermore, regions encompassing MtrH and MtrAcyt are highly flexible, limiting the local resolution. To improve resolution in these flexible regions, we performed masked 3D-classification, 3D variability analysis and local refinements, yielding

focused maps with resolutions of 2.5 Å for MtrAcyt and 3.2 Å for MtrH. Combining these maps enabled the generation of a composite map encompassing the full complex, which in turn allowed us to obtain a complete atomic model of the asymmetric Mtr complex.”

lacks several references to figure panels, for example but not limited to Fig 2A clearly indicating subunit MtrA, Fig S2 showing the processing pipeline, Fig S5d indicating the factor III. The lack of references to structures makes reading and understanding the paragraph difficult.

-On page 6-7, the paragraph

“This arrangement creates a vestibule connected to the cytoplasmic solvent, as indicated by resolved water molecules in our structure. This vestibule is closed at the border to the membrane plane by three phenylalanines (Phe210) from MtrA. In the membrane plane, the three MtrA helices bend outwards and are replaced as the core of the stalk by the MtrG helices (G44-G66), which form a three-helix bundle along the threefold axis. This bundle, characterised by conserved hydrophobic residues across methanogenic species, together with Phe210 from MtrA, acts as a tight hydrophobic seal that prevents ion leakage during conformational changes within the Mtr complex. Notably, helix B is discontinuous, featuring a cytosol-protruding loop spanning residues Pro56 to Thr70, which interacts directly with the membrane-spanning subunit MtrE. Additionally, helices F and B wrap around and cross each other between residues MtrF:42–53 and MtrB:71–80, causing a pronounced tilt of MtrF away from the threefold axis. This disruption breaks local tetrameric arrangement within the transmembrane region. Within the resulting gap, a minimum of five well-defined archaeal ether lipids are bound per trimer. The density quality allowed us to model a total of 15 lipids, though additional lipids may be present. These lipids are deeply embedded in the complex, acting as integral non-protein structural components of the Mtr complex, bridging and stabilising interactions between membrane-spanning subunits.”

refers to several results without corresponding panels, namely (1) the vestibule featuring resolved water molecules, (2) the hydrophobic seal, (3) a representation of the mentioned interactions of MtrF, B, E and lipids. I suggest adding a supplementary figures showing these panels.

-Fig 3 add a panel highlighting the active sites of MtrH and the position of MtrA to understand the sentence “Such an orientation facilitates binding of the cytoplasmic domain of the methyl-group carrying MtrA. As the active sites of the MtrH dimer face in opposite directions, with only one oriented toward MtrA, it suggests that only the membrane-proximal MtrHp is functionally relevant” from page 9.

-Figure 3D-E legend: how are the sequences for the conservation analysis sourced? A description of this analysis seems to be missing in the methods.

-In Fig 4, or in a new supplementary figure, add one or more panels displaying the mentioned lipid and protein connections to clarify the sentence “The MtrCDE trimer interacts with all components of the central stalk, except for the trimeric core-forming MtrG, and are stabilized by five lipids. Within the MtrCDE trimer only MtrE directly interacts with the stalk subunits, while MtrC and MtrD are positioned on the exterior of the protein complex. Notably, transmembrane helices TM7 and TM8 of MtrD interact specifically with 2-hydroxyarchaeolphosphatidylinositol and archaeolphosphatidylethanolamine lipids.” on page 10-11

-On page 11 “Structural comparisons using DALI reveal that these proteins possess a rare fold with no close structural analogs among other membrane proteins. However, modest similarities exist: MtrE shows resemblance to the heme transporter HmuUV, the vitamin B12 import system permease protein BtuC, and, intriguingly, the sodium-translocating RnfE. Despite low sequence identity, the resemblance to RnfE raises the possibility that this fold may have evolved to mediate sodium translocation under energy-limited conditions.” the data is not shown: please add a supplementary figure showing the DALI analysis and the structural comparison with the mentioned proteins (BtuC, HmuUV, RnfE).

-Fig 5F: it would be helpful to show the Na binding site in the figure, since this is cited on page 13 “The presence of MtrI in the cavity, along with its interaction with Arg112, therefore likely occludes the CoM binding site and blocks the sodium channel.”

-On page 13, the sentence “Superimposing MtrCDE not containing MtrI with the site of the trimer showing the MtrCDE–MtrI complex reveals no major structural differences. One notable exception is a slight rearrangement of the cytosolic loop in MtrC (residues 65–71), which includes Tyr66, to accommodate MtrI binding. This loop also appears more rigid in the cryo-EM density of the MtrCDE–MtrI complex.” lacks a corresponding figure.

-On page 17, the authors refer to transient binding sites for MtrA, “Additionally, the structure implicates two possible regions where MtrA may transiently bind - one at the membrane-proximal MtrHp TIM-barrel and another at the cavity formed by the MtrCDE trimer, where a potential CoM binding site has been proposed” but it is not clear what they refer to: this should be clarified with a figure or removed from the discussion.

Related to this point, it would be good if the authors could expand their discussion on the MtrA binding: the structures shows that the cytosolic domain only binds (1) in one copy, while three copies of the protein are present as can be seen from the transmembrane domains, and (2) the binding is mediated by MtrI. The questions therefore arises whether MtrI might be required to obtain the observed conformation and what would happen in absence of MtrI, as well as where are the two missing cytosolic domains. I suppose part of the discussion on the missing domains might be included in the text cited above, but this is not very clear and I think it would be valuable for the community if the authors could discuss these points. Maybe at some point of the processing they observed low-resolution densities that can provide hypotheses in this regard?

-Fig 4: It is not clear how the authors proved that indeed Na is bound to the complex. There is no mention of biochemical

assays or literature references. This point should be clarified.

Minor comments

-On page 4, in the sentence “obtained at a resolution of 2.1 Å in C1.” clarify that C1 refers to the symmetry (I assume), as this is probably not understandable to non-structural biologists.

-On page 5 “multi-spanning” and “single-spanning” should be preceded by “transmembrane” for clarity.

-Fig 2c legend add reference to the lipids: supposedly the position of several lipids is indicated as a purple stripe?

-In Fig 4A, it would help to color MtrABFG in grey to highlight the stalk in the model and clarify the sentence “Three integral membrane MtrCDE trimers are attached symmetrically around the central MtrABFG stalk” on page 10.

-Also, in the legend of Fig 4 add that the Na ion is shown as purple ball, this is missing.

-On page 13, the sentence “stabilized by a series of charged and aromatic interactions (Fig. 5D)” refers to figure 5G, not 5D.

-The sentence

“When comparing Methanosarcinales with other methanogenic orders, two major differences stand out: (i) Methanosarcinales can grow on additional substrates, facilitated by the presence of methanophenazines and additional energy-conserving systems that directly affect Mtr function, which e.g. is reversed during methylotrophic growth (methanol, methylamines), and (ii) Methanosarcinales often inhabit oxygen-rich environments.” lacks references.

Version 1:

Reviewer comments:

Reviewer #1

(Remarks to the Author)

The authors have appropriately addressed my prior review. Here a few suggestions how to further strengthen the new text in the introduction:

L51: starting of sentence is weird – rephrase to “They produce...”?

L54: citations to this statement should include more papers on non-methanogenic methane production, e.g. methane from nitrogenase, as well as from methylamines (Wang..McDermott et al 2021)

L76: why methanol and not methylated compounds to be more inclusive?

L82: citation to support this statement?

Reviewer #2

(Remarks to the Author)

I thank the authors for carefully addressing the comments, I believe the manuscript is now greatly improved and only have the following residual minor comments.

- I recommend adding the clarification about the sample purification included in the rebuttal to the Methods section, on page 26/line 672. The text from the rebuttal is copied below: references to figures should be added to this text.

No size-exclusion chromatography was performed with the LMNG-solubilised sample prior to Cryo-EM sample preparation, as the StrepTactin purification already yielded a highly pure preparation. LMNG is known to bind membrane proteins strongly and can be used at sub-CMC concentrations, which prevents the formation of empty detergent micelles. Therefore, a further SEC step after affinity purification was not required. We further confirmed the integrity, homogeneity, and stability of the complex for cryo-EM analysis using mass photometry.

- In Fig S8, I recommend pointing to Arg 112 in the figure. While it is the only positively charged residue shown, an explicit indication will help the non-specialist reader. In the same figure, I recommend specifying that sodium is shown as a pink sphere, in the legend.

-In page 15, line 368, I recommend adding the precise and exhaustive explanation behind the sodium density assignment to the text. while some elements are present already, part is missing and I think it would be helpful to add them to the text. I copy below the text from the rebuttal that I recommend including in the manuscript, after line 368.

Furthermore, the second step of the Mtr-catalyzed reaction has been demonstrated to be sodium dependent, with reaction rates increasing at higher Na⁺ concentrations and an estimated K_d for sodium binding of approximately 50 μM (Weiss, Gärtner and Thauer, 1994). Also, samples contained 150 mM NaCl, ensuring saturation in binding under our experimental conditions. Finally, the residues coordinating the ion are highly conserved among Mtr homologs, further supporting the assignment of Na⁺ at this site.

- In the reporting summary, the description and validation of antibodies is missing.

REVIEWER COMMENTS

Reviewer #1 (Remarks to the Author):

Tristan Reif-Trauttmansdorff et al. describe in their manuscript the cryoEM structure of the methanogenic Na⁺ translocating methyl transferase complex Mtr. They report on the overall structure and possible catalytic mechanism including a previously unrecognized Na⁺ binding site. In the course of their characterization, they discover a small regulatory protein MtrI present only in Methanosarcinales, possibly aiding against oxidative stress.

Overall, the authors report novel findings on an important enzyme of the methanogenesis pathway. They build on work done previously, both in sequence-based and structure-based investigations and add new pieces to the puzzle.

I have two main parts I'm critical about. The first is the way that the authors express their statements. They are often very bold and not refined to appropriately reflect the more nuanced state of the art. For non-experts reading this paper this can lead to severe misunderstandings of the field so I urge the authors to correct this. In my detailed comments I've named them where they appear. The introduction is also scarce in use of literature.

We thank the reviewer for their thoughtful and constructive evaluation of the manuscript and for recognizing its novel findings. We appreciate the concern regarding the tone and presentation of certain statements in the manuscript. In response, we have carefully revised the text to adopt a more nuanced and precise wording that better reflects the current state of knowledge and included additional references.

The second critique is directed at their analyses and statements surrounding the phylogenetic distribution of mtrI. The authors neither report how they determine that mtrI is restricted to Methanosarcinales (sequence or structure based search? Both need to be done. Cutoffs? Methods?) nor do they show any data that all Methanosarcinales carry mtrI. This analysis must go beyond a quick BLAST search but rather encompass comparisons of genome content on a larger scale. If orfs are not appropriately annotated for small such small proteins they may be missed by simple database searches, leading to faulty reporting; primary data genome comparisons may be needed. In order to satisfy this criticism, new and extensive *in silico* analysis needs to be performed currently not reported on in the manuscript.

We thank the reviewer for their valuable criticism. We have now performed and included a detailed *in silico* analysis. The revised manuscript now includes a detailed description of the BLAST search strategy and applied cutoffs in the Materials and Methods section. The results are presented in Supplementary Figures S10 and S11 and Supplementary Tables 3 and 4. Please also see our responses to the related comments below.

Line numbers would have been helpful to facilitate the review process.

We thank the reviewer for the observation regarding the missing line numbers. We have now added line numbers throughout the manuscript.

Abstract first sentence: contribute reads strange, emit?

Indeed, thank you. We made the change accordingly.

Introduction

Anaerobic conditions should be anoxic conditions. Metabolism is anaerobic, conditions are anoxic.

We thank the reviewer for pointing this out, we have corrected that throughout the manuscript.

The represent the only organisms... while I understand the notion of the author's, there are numerous other metabolisms that produce methane in trace amounts, e.g. demethylation reactions of methylphosphonates. The sentence as is does not represent the current state of the field. Please rephrase the sentence carefully.

We thank the reviewer for clarifying this point. We rephrased the respective part in a more nuanced, state of the art, manner.

Why is 'greenhouse gas' in brackets? I would replace climate relevant gas with greenhouse gas

We thank the reviewer for the suggestion. We have replaced "climate relevant gas" with "greenhouse gas" in the text to improve clarity.

There are several, phylogenetically not related pathways: I disagree with this sentence. Of course they are phylogenetically related, methanogenesis was an ancestral trait of the last common ancestor of methanogens. I urge the authors to phrase their statements more carefully.

We very much thank the reviewer for their valuable comment. We have made the corresponding changes in the revised manuscript to improve accuracy.

This sodium-motive force drives chemiosmotic energy conservation and ATP synthesis: please add appropriate citations for this statement

Recent advances in ribosome profiling...: it would be nice to see some literature references underpinning these statements.

We are grateful for the constructive suggestion. We have made the corresponding changes in the revised manuscript by adding the requested references.

Results

Methanosarcina in italics

Thank you, this has been updated in the manuscript.

McrB promotor should be *mcrB* promotor (*mcrB* in italics)

We appreciate your careful reading and have fixed the typo.

Of a trimer of a: remove a

Thanks for your suggestion; we have addressed it in the revised version.

Highlighting the conservation of this binding sites in all methanogens: by comparing two structures one cannot conclude that this is conserved in all methanogens. Please phrase more carefully.

We thank the reviewer for pointing this out. We agree that conservation across all methanogens cannot be concluded from only two structures. We have revised the sentence to more accurately reflect the available evidence.

MtrI apparently coordinates the cobalt atom of FIII; this seems like an essential function to me while MtrI is constrained to Methanosarcinales. In my opinion the authors need to mention how they think non-Methanosarcinales Mtr coordinates the cobalt atom of FIII if MtrI is absent

We thank the reviewer for raising this important point. In our structure, the cobalt atom of FIII is coordinated by two axial ligands: His85 of MtrA as the lower ligand and Cys40 of MtrI as the upper ligand. This coordination pattern suggests a Co(III) oxidation state. Our experimental data show that MtrI binding occurs only after oxygen exposure, indicating that this interaction is not part of the catalytic cycle but rather a regulatory response. In line with this, MtrI also blocks the central cavity and putatively the sodium channel, thereby arresting MtrA movement and likely stalling ion flow under oxidative conditions. The physiological consequence of this mechanism remains unclear; it may function as a degradation signal and/or as a plug against sodium leakage, but we currently lack direct evidence.

Thus, in Methanosarcinales, MtrI most likely acts as an oxygen-dependent regulatory element rather than a universally essential component of Mtr function. In species lacking MtrI, coordination of FIII and regulation of MtrA may be mediated by different, as yet unidentified, factors. During the catalytic cycle itself, the cobalt atom alternates between being unligated in its Co(I) state and coordinated by two axial ligands in its methylated Co(III) state, with a histidine as the lower ligand and the methyl group as the upper ligand.

Discussion

...with a core bioenergetic complex in Methanoarchaea: no, not in Methanoarchaea. In Methanosarcinales.

We thank the reviewer for the comment. While our structural study was performed in *Methanosarcina*, to our knowledge this represents the first structural evidence of a small protein associated with a core bioenergetic complex in any methanogenic archaea (Methanoarchaea). We have revised the sentence for clarity.

I think the authors very quickly conclude that MtrI is absent in methanogens outside of Methanosarcinales. I couldn't find in their methods how they checked. Only sequence based BLAST analysis (threshold for identity?), or also structure-based similarity searches? Both are

required, and thresholds must be provided, defining presence/absence of homologous proteins. As the authors included ANMEs in their introduction I also expect to read whether ANME harbor mtrI.

We thank the reviewer for pointing out this fundamental point. We have included several BLAST (e.g. tBLASTn against all available euryarchaeal genomes as of July 2024), as well as Foldseek searches. Furthermore, we have included the requested information on workflow and cut-offs in materials and methods, as well as result tables and supplementary figures (Supp. Data Table 3, 4 and Supp. Fig. S10,11). We have found MtrI to be exclusively, but not universally, present within Methanosarcinales (suppl. Fig. S10). Furthermore, several ANME-2 archaea harbour MtrI, including *Methanoperedens* sp. and *Methanocomedens* sp.

The authors state that Methanosarcinales harbor mtrI. Have they checked that all Methanosarcinales contain them? This is not a quick analysis but I think is required if the authors would like to state that all Methanosarcinales have them. The way how they check this needs to be described, too.

We thank the reviewer for this comment and agree. Although we did not specifically state MtrI was present in all Methanosarcinales, we now clarified this point – MtrI was not found in all Methanosarcinales genomes and seems restricted to the families Methanocomedenaceae, Methanoperedenaceae and Methanosarcinaceae, whilst not being present in every genera or species. Detailed MtrI distribution can now also be observed in Supp. Fig. S10.

I was also disappointed that the authors did not take available datasets on transcriptomes and proteomes to search for expression/presence of MtrI in those datasets.

Thank you for highlighting this. We have incorporated the requested information in the updated version.

Materials:

How were *E. coli* buffers made anoxic? By addition reducing agents? Please add to both *E. coli* paragraphs

We thank the reviewer for pointing this out. We have updated the *E. coli* paragraphs accordingly.

How much sample was used for ICP-MS?

We thank the reviewer for the question. The sample volume used (40 μ L at 0.09 mg/mL protein) is now explicitly stated in the Methods section.

Reviewer #2 (Remarks to the Author):

Summary

The manuscript describes the structure of the Mtr complex, a methyl-transferase from methanogenic archaea. Based on the structure, the authors identify a novel complex component and propose an oxygen-stress response mechanism.

The results are relevant to bioenergetics, structural biology and from a broad evolutionary perspective and the data is solid.

However a significant amount of the described results is not shown in figures. This makes comprehension of the manuscript difficult. I therefore recommend that the authors expand the manuscript by including the missing representations, in addition to addressing minor points listed below.

In essence, the results are noteworthy, novel and of significance for a broad audience across multiple disciplines and the methodology is sound. However, for the sake of clarity and reproducibility all the results that are described and discussed should be shown and this is currently missing in some parts of the manuscript. I therefore recommend the article is revised and updated accordingly.

- We thank the reviewer for their careful and constructive evaluation of our manuscript. We also appreciate the emphasis on showing all described results for clarity and reproducibility. In response, we have added all previously missing representations. These are included in both the main figures and the supplementary materials. This way, every result discussed is now visually represented and clearly illustrated.

Main comments

-For resubmission, line and page numbers should be added.

- We thank the reviewer for pointing this out and added line and page numbers.

-On page 2, the meaning of the sentence "In methylotrophic methanogens that disproportionate methanol into methane and CO₂, this reaction is reversed" is not clear. Particularly, it is not clear what "disproportionate" means in this sentence.

- We thank the reviewer for his comment. Disproportionation of methanol refers to the fact, that a part of methanol (25%) is oxidized to generate the reducing equivalents for the reduction of methanol (75%) We have now clarified the sentence to explain the meaning of "disproportionation" in this context.

-Also on the same page, the sentence "anaerobic methanotrophic archaea (ANME) oxidize methane to CO₂ operate the methanogenetic pathway and thus the Mtr enzyme entirely in reverse." is unclear. Maybe punctuation is missing?

- We thank the reviewer for pointing this out. We have revised the sentence for clarity: "Similarly, anaerobic methanotrophic archaea (ANME) oxidize methane to CO₂ by operating the methanogenesis pathway, and thus the Mtr enzyme, entirely in reverse."

-Fig S1c should include the SEC profile of the LMNG-solubilised sample, since this is the sample used for structure determination, instead of the DDM-solubilised material.

- We thank the reviewer for his helpful comment. No size-exclusion chromatography was performed with the LMNG-solubilised sample prior to Cryo-EM sample preparation, as the StrepTactin purification already yielded a highly pure preparation. LMNG is known to bind membrane proteins strongly and can be used at sub-CMC concentrations, which prevents the formation of empty detergent micelles. Therefore, a further SEC step after affinity purification was not required. We further confirmed the integrity, homogeneity, and stability of the complex for cryo-EM analysis using mass photometry.

-In Fig 1, indicate (1) the membrane in the side views of the map and model, as well as (2) both of the soluble compartments on the two sides of the membrane.

- We thank the reviewer for pointing this out. Figure 1 and its legend have been updated: in the side views of both the map and the model, the membrane is indicated, and both the cytosol and the extracellular site are labelled.

-Also in Fig 1 add a panel highlighting the position of the active sites and indication of the reactions (substrates and products), to clarify which reactions happen where in the complex.

- We acknowledge the reviewers comment and added two more panels. One depicting the Mtr reaction equation with the substrate and product structural formulas shown. The second showing the Mt- complex in surface view and the location where the partial reactions are happening.

-On page 5, the paragraph

"One of the stalk subunits, MtrA, features a cytoplasmic corrinoid-binding domain, MtrA_{cyt}, connected via a long, flexible linker. This domain harbours the 5-hydroxybenzimidazolylcobamide cofactor (Factor III), that serves as mobile carrier element in the methyltransferase reaction. Our cryo-EM analysis revealed that only a subset of particles exhibited a well-resolved MtrA_{cyt} domain bound to a single MtrCDE subcomplex, resulting in an overall asymmetry of the complex. Furthermore, regions encompassing MtrH and MtrA_{cyt} are highly flexible, limiting the local resolution. To improve resolution in these flexible regions, we performed masked 3D-classification, 3D variability analysis and local refinements, yielding focused maps with resolutions of 2.5 Å for MtrA_{cyt} and 3.2 Å for MtrH. Combining these maps enabled the generation of a composite map encompassing the full complex, which in turn allowed us to obtain a complete atomic model of the asymmetric Mtr complex."

lacks several references to figure panels, for example but not limited to Fig 2A clearly indicating subunit MtrA, Fig S2 showing the processing pipeline, Fig S5d indicating the factor III. The lack of references to structures makes reading and understanding the paragraph difficult.

- We thank the reviewer for pointing this out. We edited the paragraph to include additional references to the relevant figure panels. While MtrA_{cyt} and its interaction with MtrI/MtrCDE is described in more detail in a later result section, the introductory description, in addition to Fig.1, now also refers to the cryo-EM processing tree (Supp. Fig. S2c) and to the figure gallery, detailing the newly added Factor III structural formula (Supp. Fig. S14e).

-On page 6-7, the paragraph

"This arrangement creates a vestibule connected to the cytoplasmic solvent, as indicated by resolved water molecules in our structure. This vestibule is closed at the border to the membrane plane by three phenylalanines (Phe210) from MtrA. In the membrane plane, the three MtrA helices bend outwards and are replaced as the core of the stalk by the MtrG helices (G44-G66), which form a three-helix bundle along the threefold axis. This bundle, characterised by conserved hydrophobic residues across methanogenic species, together with Phe210 from MtrA, acts as a tight hydrophobic seal that prevents ion leakage during conformational changes within the Mtr complex. Notably, helix B is discontinuous, featuring a cytosol-protruding loop spanning residues Pro56 to Thr70, which interacts directly with the membrane-spanning subunit MtrE. Additionally, helices F and B wrap around and cross each other between residues MtrF:42–53 and MtrB:71–80, causing a pronounced tilt of MtrF away from the threefold axis. This disruption breaks local tetrameric arrangement within the transmembrane region. Within the resulting gap, a minimum of five well-defined archaeal ether lipids are bound per trimer. The density quality allowed us to model a total of 15 lipids, though additional lipids may be present. These lipids are deeply embedded in the complex, acting as integral non-protein structural components of the Mtr complex, bridging and stabilising interactions between membrane-spanning subunits."

refers to several results without corresponding panels, namely (1) the vestibule featuring resolved water molecules, (2) the hydrophobic seal, (3) a representation of the mentioned interactions of MtrF, B, E and lipids. I suggest adding a supplementary figures showing these panels.

- We thank the reviewer for pointing out this shortcoming. We added a supplementary figure (Supp. Fig. S3) that provides a detailed view of the core of the stalk, including (i) the water-filled vestibule in the cytosolic region and (ii) the hydrophobic seal in the membrane. In addition, we added a supplementary figure (Supp. Fig. S4) that shows the interactions between the MtrCDE trimer, the stalk subunits, and the stabilizing lipids (see also our reply to the third question below).

-Fig 3 add a panel highlighting the active sites of MtrH and the position of MtrA to understand the sentence "Such an orientation facilitates binding of the cytoplasmic domain of the methyl-group carrying MtrA. As the active sites of the MtrH dimer face in opposite directions, with only one oriented toward MtrA, it suggests that only the membrane-proximal MtrHp is functionally relevant" from page 9.

- We thank the reviewer for the suggestion. While we do not have experimental density showing MtrA bound to MtrH, we have added a panel to Fig. 1D where the active site of the proximal MtrH, that we propose is the primarily functional MtrH of the dimer, is indicated by the reaction arrows. We also revised the text to state that, while the membrane-proximal MtrH is likely the main site for MtrA binding, an interaction with the membrane-distal MtrH cannot be excluded. In addition, we added Supplementary Fig. S13, illustrating the position and movement of the cytosolic MtrA domain between MtrHp and the MtrCDE trimer.

-Figure 3D-E legend: how are the sequences for the conservation analysis sourced? A description of this analysis seems to be missing in the methods.

- We thank the reviewer for pointing this out. We have now added a dedicated "Sequence conservation analysis" section to the Methods, detailing the species, accession numbers, alignment procedure (Clustal Omega), and logo generation (WebLogo 3.0).

-In Fig 4, or in a new supplementary figure, add one or more panels displaying the mentioned lipid and protein connections to clarify the sentence "The MtrCDE trimer interacts with all components of the central stalk, except for the trimeric core-forming MtrG, and are stabilized by five lipids. Within the MtrCDE trimer only MtrE directly interacts with the stalk subunits, while MtrC and MtrD are positioned on the exterior of the protein complex. Notably, transmembrane helices TM7 and TM8 of MtrD interact specifically with 2-hydroxyarchaeolphosphatidylinositol and archaeolphosphatidylethanolamine lipids." on page 10-11

We thank the reviewer for this helpful suggestion. As outlined in our response to a previous comment, we have added a supplementary figure that illustrates in detail the interactions between the MtrCDE trimer, the stalk subunits, and the stabilizing lipids. In addition, the corresponding text section was substantially revised to more accurately describe this complex interface. The revised paragraph now provides a spatial description of how the lipids and stalk components interact with MtrE and MtrC in a clockwise arrangement around the trimer–stalk contact region. We also corrected the assignment of the lipid interactions, clarifying that it is MtrC (not MtrD) transmembrane helices TM7 and TM8 that interact specifically with 2-hydroxyarchaeolphosphatidylinositol and archaeolphosphate lipids. The figure legend and text passage were updated accordingly.

-On page 11 "Structural comparisons using DALI reveal that these proteins possess a rare fold with no close structural analogs among other membrane proteins. However, modest similarities exist: MtrE shows resemblance to the heme transporter HmuUV, the vitamin B12 import system permease protein BtuC, and, intriguingly, the sodium-translocating RnfE. Despite low sequence identity, the resemblance to RnfE raises the possibility that this fold may have evolved to mediate sodium translocation under energy-limited conditions." the data is not shown: please add a supplementary figure showing the DALI analysis and the structural comparison with the mentioned proteins (BtuC, HmuUV, RnfE).

We thank the reviewer for this helpful suggestion. We added a supplementary figure (Supp. Fig. S4) showing the superimpositions of MtrE with representative membrane transport proteins identified in our updated DALI search (NqrE, PsaC, and BhuU), including the RMSD values reported by DALI. In addition we provided a supplementary table (Supp. Data Table S2) listing the top DALI hits (PDB50 search). The corresponding text passage has been revised for clarity. In our updated DALI analysis, the previously mentioned HmuUV and vitamin B₁₂ transporter BtuC did not appear among the top structural hits and were therefore removed from the text.

-Fig 5F: it would be helpful to show the Na binding site in the figure, since this is cited on page 13 "The presence of MtrI in the cavity, along with its interaction with Arg112, therefore likely occludes the CoM binding site and blocks the sodium channel."

We thank the reviewer for this valuable suggestion. In the original viewpoint shown in Fig. 5F, the sodium binding site is clipped and therefore not visible. To address this, we have added a supplementary figure (Supp. Fig. S8) showing the sodium binding site from an alternative viewing angle. In this representation, MtrI and MtrE are shown in cartoon with the sodium ion and coordinating residues highlighted, while MtrC and MtrD are displayed as transparent surfaces. This view illustrates how the C-terminus of MtrI extends toward the sodium pocket and interacts with Arg112 located within the cavity.

-On page 13, the sentence "Superimposing MtrCDE not containing MtrI with the site of the trimer showing the MtrCDE–MtrI complex reveals no major structural differences. One notable exception is a slight rearrangement of the cytosolic loop in MtrC (residues 65–71), which includes Tyr66, to accommodate MtrI binding. This loop also appears more rigid in the cryo-EM density of the MtrCDE–MtrI complex." lacks a corresponding figure.

- We thank the reviewer for pointing this out. We have added a new supplementary figure (Supp. Fig. S7) showing the structural comparison of MtrCDE with and without MtrI bound. The figure includes a superimposition highlighting overall similarity and a close-up of the MtrC loop (residues 65–71) accommodating MtrI binding.

-On page 17, the authors refer to transient binding sites for MtrA, "Additionally, the structure implicates two possible regions where MtrA may transiently bind - one at the membrane-proximal MtrHp TIM-barrel and another at the cavity formed by the MtrCDE trimer, where a potential CoM binding site has been proposed" but it is not clear what they refer to: this should be clarified with a figure or removed from the discussion.

Related to this point, it would be good if the authors could expand their discussion on the MtrA binding: the structures show that the cytosolic domain only binds (1) in one copy, while three copies of the protein are present as can be seen from the transmembrane domains, and (2) the binding is mediated by MtrI. The questions therefore arise whether MtrI might be required to obtain the observed conformation and what would happen in absence of MtrI, as well as where are the two missing cytosolic domains. I suppose part of the discussion on the missing domains might be included in the text cited above, but this is not very clear and I think it would be valuable for the community if the authors could discuss these points. Maybe at some point of the processing they observed low-resolution densities that can provide hypotheses in this regard?

- We thank the reviewer for this helpful comment. We have clarified the section referring to potential transient binding sites of MtrA and added Supplementary Fig. S13 to illustrate the proposed positions of the cytosolic MtrA domain relative to the membrane-proximal MtrH (MtrHp) and the cavity formed by the MtrCDE trimer. While the current cryo-EM data do not resolve MtrA bound at these positions, the locations are supported by structural context,

previously proposed CoM binding sites, and biochemical considerations. We are currently collecting additional cryo-EM data to further characterize these transient MtrA positions experimentally.

- Regarding the second point: We have expanded the discussion to clarify the presence of the other two “missing” copies of MtrA. Examination of the consensus map at low threshold and after Gaussian filtering revealed additional, low-occupancy density corresponding to MtrA bound to the remaining MtrCDE sites via MtrI (Supp. Fig. S12). Since the consensus map was generated from 3D classification using a mask focused on one MtrA-bound site (see Supp. Fig. S2, cryo-EM processing), only one copy of MtrA appears in a well-defined, stoichiometric conformation. The weaker densities at the other sites indicate partial or transient MtrA binding, likely reflecting the absence of MtrI at these positions, which appears to render MtrA flexible or unbound under our experimental conditions.

-Fig 4: It is not clear how the authors proved that indeed Na is bound to the complex. There is no mention of biochemical assays or literature references. This point should be clarified.

- We thank the reviewer for pointing out this lack of clarification. The presence of a sodium ion was inferred based on the geometry, coordination number, and interatomic distances observed in the cryo-EM map, all of which are consistent with Na⁺ coordination. The ion is octahedrally coordinated by six oxygen ligands with Na–O distances ranging from 1.9 to 2.7 Å, in line with characteristic Na⁺-binding geometries described in protein crystal structures (Nayal and Cera, 1996). The corresponding density is markedly stronger than that of surrounding water molecules, as expected at this resolution. Additionally, a recent cryo-EM structure of the Mtr complex from *Methanothermobacter marburgensis* identifies the same conserved sodium-binding pocket, supporting our assignment of Na⁺. Furthermore, the second step of the Mtr-catalyzed reaction has been demonstrated to be sodium dependent, with reaction rates increasing at higher Na⁺ concentrations and an estimated K_d for sodium binding of approximately 50 μM (Weiss, Gärtner and Thauer, 1994). Also, samples contained 150 mM NaCl, ensuring saturation in binding under our experimental conditions. Finally, the residues coordinating the ion are highly conserved among Mtr homologs, further supporting the assignment of Na⁺ at this site.

Minor comments

-On page 4, in the sentence "obtained at a resolution of 2.1 Å in C1." clarify that C1 refers to the symmetry (I assume), as this is probably not understandable to non-structural biologists.

- We thank the reviewer for pointing this out. We have clarified in the text that C1 refers to the symmetry applied during reconstruction.

-On page 5 "multi-spanning" and "single-spanning" should be preceded by "transmembrane" for clarity.

- We thank the reviewer for pointing out this ambiguity. We have corrected it by specifying "multi-spanning transmembrane subunits" and "single-spanning transmembrane subunits".

-Fig 2c legend add reference to the lipids: supposedly the position of several lipids is indicated as a purple stripe?

- We thank the reviewer for this helpful suggestion. We have revised the legend of Fig. 2C accordingly.

-In Fig 4A, it would help to color MtrABFG in grey to highlight the stalk in the model and clarify the sentence "Three integral membrane MtrCDE trimers are attached symmetrically around the central MtrABFG stalk" on page 10.

We thank the reviewer for this helpful suggestion. To maintain the overview nature of Fig. 4A, we kept the original color scheme unchanged. However, to clarify the interactions between the central MtrABFG stalk, the lipids and the three symmetrically attached MtrCDE trimers, we have added a new supplementary figure (Supp. Fig. S4) that highlights these structural relationships in detail.

-Also, in the legend of Fig 4 add that the Na ion is shown as purple ball, this is missing.

- We thank the reviewer for this suggestion. The figure legend for Fig. 4 has been updated to indicate that sodium ions are represented as purple spheres

-On page 13, the sentence "stabilized by a series of charged and aromatic interactions (Fig. 5D)" refers to figure 5G, not 5D.

- We thank the reviewer for pointing this out. The reference has been corrected so that the sentence now cites Fig. 5G instead of Fig. 5D.

-The sentence

"When comparing Methanosarcinales with other methanogenic orders, two major differences stand out: (i) Methanosarcinales can grow on additional substrates, facilitated by the presence of methanophenazines and additional energy-conserving systems that directly affect Mtr function, which e.g. is reversed during methylotrophic growth (methanol, methylamines), and (ii) Methanosarcinales often inhabit oxygen-rich environments."

lacks references.

- We thank the reviewer for pointing this out. We have added references detailing the biochemical and energetic differences between Methanosarcinales and other methanogenic orders (Deppenmeier and Müller, 2008) as well as studies addressing the oxygen tolerance and oxidative stress responses of Methanosarcinales (Brioukhanov *et al.*, 2000; Horne and Lessner, 2013; Jasso-Chávez *et al.*, 2015)

Brioukhanov, A. *et al.* (2000) "Protection of *Methanosarcina barkeri* against oxidative stress: identification and characterization of an iron superoxide dismutase," *Archives of Microbiology*, 174(3), pp. 213–216. Available at: <https://doi.org/10.1007/s002030000180>.

Deppenmeier, U. and Müller, V. (2008) "Life Close to the Thermodynamic Limit: How Methanogenic Archaea Conserve Energy," in G. Schäfer and H.S. Penefsky (eds.) *Bioenergetics: Energy Conservation and Conversion*. Berlin, Heidelberg: Springer, pp. 123–152. Available at: https://doi.org/10.1007/400_2006_026.

Horne, A.J. and Lessner, D.J. (2013) "Assessment of the oxidant tolerance of *Methanosarcina acetivorans*," *FEMS microbiology letters*, 343(1), pp. 13–19. Available at: <https://doi.org/10.1111/1574-6968.12115>.

Jasso-Chávez, R. *et al.* (2015) "Air-Adapted *Methanosarcina acetivorans* Shows High Methane Production and Develops Resistance against Oxygen Stress," *PLOS ONE*, 10(2), p. e0117331. Available at: <https://doi.org/10.1371/journal.pone.0117331>.

Nayal, M. and Cera, E.D. (1996) "Valence Screening of Water in Protein Crystals Reveals Potential Na⁺ Binding Sites," *Journal of Molecular Biology*, 256(2), pp. 228–234. Available at: <https://doi.org/10.1006/jmbi.1996.0081>.

Weiss, D.S., Gärtner, P. and Thauer, R.K. (1994) "The Energetics and Sodium-Ion Dependence of N5-Methyltetrahydromethanopterin:Coenzyme M Methyltransferase Studied with Cob(I)Alamin as Methyl Acceptor and Methylcob(III)Alamin as Methyl Donor," *European Journal of Biochemistry*, 226(3), pp. 799–809. Available at: <https://doi.org/10.1111/j.1432-1033.1994.00799.x>.

Reviewer #1 (Remarks to the Author):

The authors have appropriately addressed my prior review. Here a few suggestions how to further strengthen the new text in the introduction:

L51: starting of sentence is weird – rephrase to “They produce...”?

We thank the reviewer for this comment. We changed the starting of the phrase accordingly.

L54: citations to this statement should include more papers on non-methanogenic methane production, e.g. methane from nitrogenase, as well as from methylamines (Wang..McDermott et al 2021)

We thank the reviewer for this comment. We added additional citations.

L76: why methanol and not methylated compounds to be more inclusive?

We thank the reviewer for pointing this out. We edited the phrase accordingly.

L82: citation to support this statement?

We thank the reviewer this comment. We added an appropriate reference.

Reviewer #2 (Remarks to the Author):

I thank the authors for carefully addressing the comments, I believe the manuscript is now greatly improved and only have the following residual minor comments.

- I recommended adding the clarification about the sample purification included in the rebuttal to the Methods section, on page 26/line 672. The text from the rebuttal is copied below: references to figures should be added to this text. No size-exclusion chromatography was performed with the LMNG-solubilised sample prior to Cryo-EM sample preparation, as the StrepTactin purification already yielded a highly pure preparation. LMNG is known to bind membrane proteins strongly and can be used at sub-CMC concentrations, which prevents the formation of empty detergent micelles. Therefore, a further SEC step after affinity purification was not required. We further confirmed the integrity, homogeneity, and stability of the complex for cryo-EM analysis using mass photometry.

We thank the reviewer this comment. We added the answer from the rebuttal to the methods section, more precisely to the “Molecular size determination” paragraph.

- In Fig S8, I recommend pointing to Arg 112 in the figure. While it is the only positively charged residue shown, an explicit indication will help the non-specialist reader. In the same figure, I recommend specifying that sodium is shown as a pink sphere, in the legend.

We thank the reviewer for this helpful suggestion. We have updated Fig. S8 to include an explicit label for Arg112, which clarifies its position. In addition, the figure legend now specifies that the sodium ion is represented as a pink sphere.

-In page 15, line 368, I recommend adding the precise and exhaustive explanation behind the sodium density assignment to the text. While some elements are present already, part is missing and I think it would be helpful to add them to the text. I copy below the text from the rebuttal that I recommend including in the manuscript, after line 368.

Furthermore, the second step of the Mtr-catalyzed reaction has been demonstrated to be sodium dependent, with reaction rates increasing at higher Na⁺ concentrations and an estimated K_d for sodium binding of approximately 50 μM (Weiss, Gärtner and Thauer, 1994). Also, samples contained 150 mM NaCl, ensuring saturation in binding under our experimental conditions. Finally, the residues coordinating the ion are highly conserved among Mtr homologs, further supporting the assignment of Na⁺ at this site.

We thank the reviewer for this helpful suggestion. We have expanded the relevant section by incorporating the full explanation provided in the rebuttal.

- In the reporting summary, the description and validation of antibodies is missing.

We thank the reviewer this comment. We added the appropriate information to the reporting summary.